# Instance-Based Uncertainty Estimation for Gradient-Boosted Regression Trees

**Jonathan Brophy**[*]
University of Oregon
jbrophy@cs.uoregon.edu

**Daniel Lowd**
University of Oregon
lowd@cs.uoregon.edu

## Abstract

Gradient-boosted regression trees (GBRTs) are hugely popular for solving tabular regression problems, but provide no estimate of uncertainty. We propose Instance-Based Uncertainty estimation for Gradient-boosted regression trees (IBUG), a simple method for extending any GBRT point predictor to produce probabilistic predictions. IBUG computes a non-parametric distribution around a prediction using the $k$-nearest training instances, where distance is measured with a tree-ensemble kernel. The runtime of IBUG depends on the number of training examples at each leaf in the ensemble, and can be improved by sampling trees or training instances. Empirically, we find that IBUG achieves similar or better performance than the previous state-of-the-art across 22 benchmark regression datasets. We also find that IBUG can achieve improved probabilistic performance by using different base GBRT models, and can more flexibly model the posterior distribution of a prediction than competing methods. We also find that previous methods suffer from poor probabilistic calibration on some datasets, which can be mitigated using a scalar factor tuned on the validation data. Source code is available at https://github.com/jjbrophy47/ibug.

## 1 Introduction

Despite the impressive success of deep learning models on unstructured data (e.g., images, audio, text), gradient-boosted trees [26] remain the preferred choice for *tabular* or *structured* data [52]. In fact, Kaggle CEO Anthony Goldbloom recently described gradient-boosted trees as the most "glaring difference" between what is used on Kaggle and what is "fashionable in academia" [34].

Our focus is on tabular data for regression tasks, which vary widely from financial [1] and retail-product forecasting [43] to weather [29, 28] and clinic-mortality prediction [6]. Gradient-boosted regression trees (GBRTs) are known to make accurate *point predictions* [42] but provide no estimate of the prediction uncertainty, which is desirable for both forecasting practitioners [9, 68] and the explainable AI (XAI) community [25, 69, 2] in general. Recently, Duan et al. [22], Sprangers et al. [64], and Malinin et al. [44] introduced NGBoost, PGBM, and CBU (CatBoost [52] with uncertainty), respectively, new gradient boosting algorithms that provide state-of-the-art probabilistic predictions. However, NGBoost tends to underperform as a point-predictor, and PGBM and CBU are limited in the types of distributions they can use to model the output.

We introduce a simple yet effective method for enabling *any* GBRT point-prediction model to produce probabilistic predictions. Our proposed approach, Instance-Based Uncertainty estimation for Gradient-boosted regression trees (*IBUG*), has two key components: 1) We leverage the fact that GBRTs accurately model the conditional mean and use this point prediction as the mean in a probabilistic forecast; and 2) We identify the $k$ training examples with the greatest *affinity* to the

---

[*]https://www.jonathanbrophy.com

36th Conference on Neural Information Processing Systems (NeurIPS 2022).

target instance and use these examples to estimate the uncertainty of the target prediction. We define the affinity between two instances as the number of times both instances appear in the same leaf throughout the ensemble. Thus, our method acts as a wrapper around any given GBRT model.

In experiments on 21 regression benchmark datasets and one synthetic dataset, we demonstrate the effectiveness of IBUG to deliver on par or improved probabilistic performance as compared to existing state-of-the-art methods while maintaining state-of-the-art point-prediction performance. We also show that probabilistic predictions can be improved by applying IBUG to different GBRT models, something that NGBoost and PGBM cannot do. Additionally, IBUG can use the training instances closest to the target example to directly model the output distribution using any parametric *or* non-parametric distribution, something NGBoost, PGBM, *and* CBU cannot do. Finally, we show that sampling trees dramatically improves runtime efficiency for computing training-example affinities without having a significant detrimental impact on the resulting probabilistic predictions, allowing IBUG to scale to larger datasets.

## 2 Notation & Background

We assume an instance space $\mathcal{X} \subseteq \mathbb{R}^p$ and target space $\mathcal{Y} \subset \mathbb{R}$. Let $\mathcal{D} := \{(x_i, y_i)\}_{i=1}^n$ be a training dataset in which each instance $x_i \in \mathcal{X}$ is a $p$-dimensional vector $(x_i^j)_{j=1}^p$ and $y_i \in \mathcal{Y}$.

### 2.1 Gradient-Boosted Regression Trees

Gradient-boosting [26] is a powerful machine-learning algorithm that iteratively adds weak learners to construct a model $f : \mathcal{X} \to \mathbb{R}$ that minimizes some empirical risk $\mathcal{L} : \mathbb{R} \times \mathbb{R} \to \mathbb{R}$. The model is defined by a recursive relationship: $f_0(x) = \gamma, \ldots, f_t(x) = f_{t-1}(x) + \eta \, m_t(x)$ in which $f_0$ is the base learner, $\gamma$ is an initial estimate, $f_t$ is the model at iteration $t$, $m_t$ is the weak learner added during iteration $t$ to improve the model, and $\eta$ is the learning rate.

Gradient-boosted regression trees (GBRTs) typically choose $\ell$ to be the mean squared error (MSE), $\gamma$ as $\frac{1}{n} \sum_{i=1}^n y_i$ (mean output of the training instances), and regression trees as weak learners. Each weak learner is typically chosen to approximate the negative gradient [44]: $m_t = \arg\min_m \frac{1}{n} \sum_{i=1}^n (-g_t^i - m(x_i))^2$ in which $g_t^i = \frac{\partial \ell(y_i, \hat{y}_i)}{\partial \hat{y}_i}$ is the functional gradient of the $i$th training instance at iteration $t$ with respect to $\hat{y}_i = f_{t-1}(x_i)$.

The weak learner at iteration $t$ recursively partitions the instance space into $M_t$ disjoint regions $\{r_t^j\}_{j=1}^{M_t}$. Each region is called a leaf, and the parameter value for leaf $j$ at tree[2] $t$ is typically determined (given a fixed structure) using a one-step Newton-estimation method [36]: $\theta_t^j = -\sum_{i \in I_t^j} g_t^i / (\sum_{i \in I_t^j} h_t^i + \lambda)$ in which $I_t^j = \{(x_i, y_i) \mid x_i \in r_t^j\}_{i=1}^n$ is the instance set of leaf $j$ for tree $t$, $h_t^i$ is the second derivative of the $i$th training instance w.r.t. $\hat{y}_i$, and $\lambda$ is a regularization constant. Thus, the output of $m_t$ can be written as follows: $m_t(x) = \sum_{j=1}^{M_t} \theta_t^j \, \mathbb{1}[x \in r_t^j]$ in which $\mathbb{1}$ is the indicator function. The final GBRT model generates a prediction for a target example $x_{te}$ by summing the values of the leaves $x_{te}$ traverses to across all $T$ iterations: $\hat{y}_{te} = \sum_{t=1}^T m_t(x_{te})$.

### 2.2 Probabilistic Regression

Our focus is on probabilistic regression—estimating the conditional probability distribution $P(y|x)$ for some target variable $y$ given some input vector $x \in \mathcal{X}$. Unfortunately, traditional GBRT models only output scalar values. Under a squared-error loss function, these scalar values can be interpreted as the conditional mean in a Gaussian distribution with some (unknown) constant variance. However, homoscedasticity is a strong assumption and unknown constant variance has little value in a probabilistic prediction; thus, in order to allow heteroscedasticity, the predicted distribution needs at least two parameters to convey both the magnitude and uncertainty of the prediction [22].

Natural Gradient Boosting (NGBoost) is a recent method by Duan et al. [22] that tackles the aforementioned problems by estimating the parameters of a desired distribution using a multi-parameter boosting approach that trains a separate ensemble for each parameter of the distribution. NGBoost employs the natural gradient to be invariant to parameterization, but requires the inversion

---

[2]We use the terms *tree* and *iteration* interchangeably.

of many small matrices (each the size of the number of parameters) to do so. Empirically, NGBoost generates state-of-the-art probabilistic predictions, but tends to underperform as a point predictor.

More recently, Sprangers et al. [64] introduced Probabilistic Gradient Boosting Machines (PGBM), a single model that optimizes for point performance, but can also generate accurate probabilistic predictions. PGBM treats leaf values as stochastic random variables, using sample statistics to model the mean and variance of each leaf value. PGBM estimates the output mean and variance of a target example using the estimated parameters of each leaf it is assigned to. The predicted mean and variance are then used as parameters in a specified distribution to generate a probabilistic prediction. PGBM has been shown to produce state-of-the-art probabilistic predictions; however, computing the necessary leaf statistics during training can be computationally expensive, especially as the number of leaves in the ensemble increases (see §5.6). Also, since only the mean and variance are predicted for a given target example, PGBM is limited to distributions using only location and scale to model the output.

Finally, Malinin et al. [44] introduce CatBoost with uncertainty (CBU), a method that estimates uncertainty using ensembles of GBRT models. Similar to NGBoost, multiple ensembles are learned to output the mean and variance. However, CBU also constructs a *virtual ensemble*—a set of overlapping partitions of the learned GBRT trees—to estimate the uncertainty of a prediction by taking the mean of the variances output from the virtual ensemble. Their approach uses a recently proposed stochastic gradient Langevin boosting algorithm [72] to sample from the true posterior via the virtual ensmble (in the limit); however, their formulation of uncertainty is limited only to the first and second moments, similar to PGBM.

To address the shortcomings of existing approaches, we introduce a simple method that performs well on both point and probabilistic performance, *and* can more flexibly model the output than previous approaches. Our method can also be applied to *any* GBRT model, adding additional flexibility.

## 3 Instance-Based Uncertainty

Instance-based methods such as $k$-nearest neighbors have been around for decades and have been useful for many different machine learning tasks [50]. However, defining neighbors based on a fixed metric like Euclidean distance may lead to suboptimal performance, especially as the dimensionality of the dataset increases. More recently, it has been shown that random forests can be used as an adaptive nearest neighbors method [18, 40] which identifies the most similar examples to a given instance using the learned model structure. This *supervised tree kernel* can more effectively measure the similarity between examples, and has been used for clustering [46] and local linear modeling [8] as well as instance-[11] and feature-based attribution explanations [51], for example.

In this work, we apply the idea of a supervised tree kernel to help model the *uncertainty* of a given GBRT prediction. Our approach, *Instance-Based Uncertainty estimation for Gradient-boosted regression trees* (IBUG), identifies the neighborhood of similar training examples to a target example using the structure of the GBRT, and then uses those instances to generate a probabilistic prediction. IBUG works for *any* GBRT, and can more flexibly model the output than competing methods.

### 3.1 Identification of High-Affinity Neighbors

At its core, IBUG uses the $k$ training examples with the largest *affinity* to the target example to model the conditional output distribution. Given a GBRT model $f$, we define the affinity between two examples simply as the number of times each instance appears in the same leaf across all trees in $f$. Thus, the affinity of the $i$th training example $x_i$ to a target example $x_{te}$ can be written as:

$$A(x_i, x_{te}) = \sum_{i=1}^{T} \mathbb{1}[R_t(x_i) = R_t(x_{te})],  \tag{1}$$

in which $R_t(x_i)$ is the leaf $x_i$ is assigned to for tree $t$. Alg. 1 summarizes the procedure for computing affinity scores for all training examples. This metric clusters similar examples together based on the learned model representation (i.e., the tree structures). Intuitively, if two examples appear in the same leaf in every tree throughout the ensemble, then both examples are predicted in an identical manner. One may also view Eq. (1) as an indication of which training examples most often affect the leaf values $x_{te}$ is assigned to and thus implicitly which examples are likely to have a big effect on the

prediction $\hat{y}_{te}$. This similarity metric is similar to the random forest kernel [18], however, unlike random forests, GBRTs are typically constructed to a shallower depth, resulting in more training examples assigned to the same leaf (see §C.5 for additional details about leaf density in GBRTs).

---

**Algorithm 1** IBUG affinity computation.

**Input:** Input instance $x \in \mathcal{X}$, GBRT model $f$.
1: **procedure** COMPUTEAFFINITIES$(x, f)$
2:     $A \leftarrow \vec{0}$            ▷ Init. train affinities
3:     **for** $t = 1 \ldots T$ **do**     ▷ Visit each tree
4:         Get instance set $I_t^l$ for leaf $l = R_t(x)$
5:         **for** $i \in I_t^l$ **do**     ▷ Increment affinities
6:             $A_i \leftarrow A_i + 1$
7: **return** $A$

**Algorithm 2** IBUG probabilistic prediction.

**Input:** Input $x \in \mathcal{X}$, GBRT model $f$, $k$ highest-affinity neighbors $A^{(k)}$, min. variance $\rho$, variance calibration parameters $\gamma$ and $\delta$, target distribution $D$.
1: **procedure** PROBPREDICT$(x, f, A^{(k)}, \rho, \gamma, \delta, D)$
2:     $\mu_{\hat{y}} \leftarrow f(x)$         ▷ GBRT scalar output
3:     $\sigma_{\hat{y}}^2 \leftarrow \max(\sigma^2(A^{(k)}), \rho)$     ▷ Ensure $\sigma^2 > 0$
4:     $\sigma_{\hat{y}}^2 \leftarrow \gamma\sigma_{\hat{y}}^2 + \delta$     ▷ Var. calibration, Eq. (2)
5: **return** $D(A^{(k)}|\mu_{\hat{y}}, \sigma_{\hat{y}}^2)$     ▷ Eq. (3)

---

### 3.2 Modeling the Output Distribution

IBUG has a multitude of choices when modeling the conditional output distribution. The simplest and most common approach is to model the output assuming a Gaussian distribution [22, 64]. We use the scalar output of $f$: $\mu_{\hat{y}_{te}} = f(x_{te})$ to model the conditional mean since GBRTs already produce accurate point predictions. Then, we use the $k$ training instances with the largest affinity to $x_{te}$—we denote this set $A^{(k)}$—to compute the variance $\sigma_{\hat{y}_{te}}^2$.

**Calibrating prediction variance.** The $k$-nearest neighbors generally do a good job of determining the relative uncertainty of different predictions, but on some datasets, the resulting variance is systematically too large or too small. To correct for this, we apply an additional affine transformation before making the prediction:

$$\sigma_{\hat{y}_{te}}^2 \leftarrow \gamma\sigma_{\hat{y}_{te}}^2 + \delta, \tag{2}$$

where $\gamma$ and $\delta$ are tuned on validation data after $k$ has been selected. Instead of exhaustively searching over all values of $\gamma$ or $\delta$, we use either the multiplicative factor (tuning $\gamma$ with $\delta = 0$) or the additive factor (tuning $\delta$ with $\gamma = 1$), and choose between them using their performance on validation data.

We find this simple calibration step consistently improves probabilistic performance for not only IBUG, but competing methods as well, and at a relatively small cost compared to training the model.

**Flexible posterior modeling.** In general, we can generate a probabilistic prediction using $\mu_{\hat{y}_{te}}$ and $\sigma_{\hat{y}_{te}}^2$ for any distribution that uses location and scale (note PGBM and CBU can *only* model these types of distributions). However, IBUG can additionally use $A^{(k)}$ to directly fit any continuous distribution $D$, including those with high-order moments:

$$\hat{D}_{te} = D(A^{(k)}|\mu_{\hat{y}_{te}}, \sigma_{\hat{y}_{te}}^2). \tag{3}$$

Eq. (3) is defined such that $D$ can be fit directly with $A^{(k)}$ using MLE (maximum likelihood estimation) [47], or may be fit using $\mu_{\hat{y}_{te}}$ or $\sigma_{\hat{y}_{te}}^2$ as fixed parameter values with $A^{(k)}$ fitting any other parameters of the distribution. Overall, directly fitting all or some additional parameters in $D$—for example, the shape parameter in a Weibull distribution—is a benefit over PGBM and CBU, which can only optimize for a *global* shape value using a gridsearch-like approach with extra validation data.

Note that NGBoost can model any parameterized distribution, but must specify this choice before training; in contrast, IBUG can optimize this choice *after* training. Additionally, IBUG may choose $D$ to be a *non-parametric density estimator* such as KDE (kernel density estimation) [62], which PGBM, CBU, and NGBoost cannot do.

### 3.3 Summary

In summary, Alg. 2 provides pseudocode for generating a probabilistic prediction with IBUG. Note Algs. 1 and 2 work for *any* GBRT model, allowing practitioners to employ IBUG to adapt multiple different point predictors into probabilistic estimators and select the model with the best performance.

Empirically, we show using different base models for IBUG can result in improved probabilistic performance than using just one (§5.3).

IBUG is a nearest neighbors approach and thus seems well-suited to estimating aleatoric uncertainty—remaining uncertainty due to irreducible error or the inherent stochasticity in the system [33]—since it can quantify the range of outcomes to be expected given the observed features. However, we use predictions on held-out data to tune the number of nearest neighbors and the variance calibration hyperparameters; thus, we effectively optimize prediction uncertainty encompassing both aleatoric uncertainty and epistemic uncertainty—error due to the imperfections of the model and the training data [20, 44]. The evaluation measures in our experiments thus also focus on predictive uncertainty.

# 4   Computational Efficiency

**Training efficiency.**   Since IBUG works with standard GBRT models, it inherits the training efficiency of modern GBRT implementations such as XGBoost [12], LightGBM [36], and CatBoost [52].

**Prediction efficiency.**   If there are $T$ trees in the ensemble and each leaf has at most $n_l$ training instances assigned to it, then IBUG's prediction time is $O(Tn_l)$, since it considers each instance in each leaf. Note training instances that do not appear in a leaf with the target instance do not increase prediction time; what matters most is thus the number of instances at each leaf. We find LightGBM often induces regression trees with large leaves—in some cases, over half the dataset is assigned to a single leaf (see §C.5 for details). Thus, prediction time still grows with the size of the dataset, as is typical for instance-based methods. This higher prediction time is the price IBUG pays for greater flexibility.

Prediction efficiency can be increased at training time by using deeper GBRTs with fewer instances in each leaf, after training by subsampling the instances considered for predictions, or at prediction time by sampling the trees used to compute affinities. We explore this last option in the next subsection.

## 4.1   Sampling Trees

The most expensive operation when generating a probabilistic prediction with IBUG is computing the affinity vector (Eq. 1). In order to increase prediction efficiency, we can instead work with a subset of the trees $\tau < T$ in the ensemble. We can build this subset by sampling trees uniformly at random, taking the first trees learned (representing the largest gradient steps), or the last trees learned (representing the fine-tuning steps).

By sampling trees, the runtime complexity reduces to $O(\tau n_l)$, which provides significant speedups when $\tau \ll T$. In our empirical evaluation, we find that taking a subset of the first trees learned generally works best, significantly increasing prediction efficiency while maintaining accurate probabilistic predictions (§5.6).

## 4.2   Accelerated $k$ Tuning

Choosing an appropriate value of $k$ is critical for generating accurate probabilistic predictions in IBUG. Thus, we aim to tune $k$ using a held-out validation dataset $\mathcal{D}_{val} \subset \mathcal{D}$ and an appropriate probabilistic scoring metric such as negative log likelihood (NLL). Unfortunately, typical tuning procedures would result in the same affinity vectors being computed—an expensive operation—for each candidate value of $k$. To mitigate this issue, we perform a custom tuning procedure that reuses computed affinity vectors for all values of $k$. More specifically, IBUG computes an affinity vector $A$ for a given validation example $x_{val}$, and then sorts $A$ in descending order (i.e., largest affinity first). Then, IBUG takes the top $k$ training instances, and generates and scores the resulting probabilistic prediction. For each subsequent value of $k$, the same sorted affinity list can be used, avoiding duplicate computation. We summarize this procedure in Alg. 3.

Once $k$ is chosen, we may encounter a new unseen target instance in which the variance of the $k$-highest affinity training examples for that target example is zero or extremely small. In this case, we set the predicted target variance to $\rho$, which is set during tuning to the minimum (nonzero) variance computed over all predictions in the validation set for the chosen $k$. In practice, we find instances of abnormally low variance to be rare with appropriately chosen values of $k$.

---

**Algorithm 3** IBUG accelerated tuning of $k$.

---

**Input:** Validation dataset $\mathcal{D}_{val} \subset \mathcal{D}$, GBRT model $f$, list of candidates $K$, target distribution $D$, probabilistic scoring metric $V$, minimum variance $\rho = 1\mathrm{e}{-}15$.

1: **procedure** FASTTUNEK($\mathcal{D}_{val}, f, K, D, V, \rho$)
2:     **for** $(x_j, y_j) \in \mathcal{D}_{val}$ **do**
3:         $A \leftarrow$ COMPUTEAFFINITIES($x_j, f$)                       ▷ Algorithm 1
4:         $A \leftarrow$ Argsort $A$ in descending order
5:         **for** $k \in K$ **do**                 ▷ Use same ordering for each $k$
6:             $A^{(k)} \leftarrow$ Take first $k$ training instances($A, k$)
7:             $\hat{D}^k_{y_j} \leftarrow$ PROBPREDICT($x_j, f, A^{(k)}, \rho, 1, 0, D$)        ▷ Algorithm 2
8:             $S^k_j \leftarrow V(y_j, \hat{D}^k_{y_j})$                ▷ Save validation score
9:     $k \leftarrow$ Select best $k$ from $S$
10:    $\rho \leftarrow$ Select minimum $\sigma^2$ from $\hat{D}^k$
11: **return** $k, \rho$

---

## 5 Experiments

In this section, we demonstrate IBUG's ability to produce competitive probabilistic and point predictions as compared to current state-of-the-art methods on a large set of regression datasets (§5.1, §5.2). Then, we show that IBUG can use different base models to improve probabilistic performance (§5.3), flexibly model the posterior distribution (§5.4), and use approximations to speed up probabilistic predictions while maintaining competitive performance (§5.6).

**Implementation and Reproducibility.** We implement IBUG in Python, using Cython—a Python package allowing the development of C extensions—to store a unified representation of the model structure. IBUG supports all modern gradient boosting frameworks including XGBoost [12], Light-GBM [36], and CatBoost [52]. Experiments are run on publicly available datasets using an Intel(R) Xeon(R) CPU E5-2690 v4 @ 2.6GHz with 60GB of RAM @ 2.4GHz. Links to all data sources as well as the code for IBUG and all experiments is available at `https://github.com/jjbrophy47/ibug`.

### 5.1 Methodology

We now compare IBUG's probabilistic and point predictions to NGBoost [22], PGBM [64], and CBU [44] on 21 benchmark regression datasets and one synthetic dataset. Additional dataset details are in §B.1.

**Metrics.** We compute the average continuous ranked probability score (CRPS ↓) and negative log likelihood (NLL ↓) [29, 76] over the test set to evaluate probabilistic performance. To evaluate point performance, we use root mean squared error (RMSE ↓). For all metrics, lower is better. See §B for detailed descriptions.

**Protocol.** We follow a similar protocol to Sprangers et al. [64] and Duan et al. [22]. We use 10-fold cross-validation to create 10 90/10 train/test folds for each dataset. For each fold, the 90% training set is randomly split into an 80/20 train/validation set to tune any hyperparameters. Once the hyperparameters are tuned, the model is retrained using the entire 90% training set. For probabilistic predictions, a normal distribution is used to model the output.

**Significance Testing.** We report counts of the number of datasets in which a given method performed better ("Win"), worse ("Loss"), or not statistically different ("Tie") relative to a comparator using a two-sided paired t-test over the 10 random folds with a significance level of 0.05.

**Hyperparameters.** We tune NGBoost the same way as in Duan et al. [22]. Since PGBM, CBU, and IBUG optimize a point prediction metric, we tune their hyperparameters similarly. We also tune variance calibration parameters $\gamma$ and $\delta$ for each method (§3.2). Exact hyperparameter values evaluated and selected are in §B.2. Unless specified otherwise, we use CatBoost [52] as the base model for IBUG.

Table 1: Probabilistic (CRPS) performance for each method on each dataset. Lower is better. Normal distributions are used for all probabilistic predictions. Results are averaged over 10 folds, and standard errors are shown in subscripted parentheses. The best method for each dataset is bolded, as well as those with standard errors that overlap the best method. *Bottom row*: Head-to-head comparison between IBUG/IBUG+CBU and each method showing the number of wins, ties, and losses (W-T-L) across all datasets. On average, IBUG+CBU provides the most accurate probabilistic predictions.

| Dataset | NGBoost | PGBM | CBU | IBUG | IBUG+CBU |
|---|---|---|---|---|---|
| Ames | $38346_{(547)}$ | $10872_{(355)}$ | $11008_{(330)}$ | $\mathbf{10434}_{(367)}$ | $\mathbf{10194}_{(368)}$ |
| Bike | $12.4_{(0.955)}$ | $1.183_{(0.041)}$ | $0.833_{(0.036)}$ | $0.974_{(0.048)}$ | $\mathbf{0.766}_{(0.032)}$ |
| California | $1e11_{(1e11)}$ | $0.222_{(0.001)}$ | $0.217_{(0.001)}$ | $0.213_{(0.001)}$ | $\mathbf{0.207}_{(0.001)}$ |
| Communities | $0.068_{(0.002)}$ | $0.068_{(0.002)}$ | $0.067_{(0.002)}$ | $\mathbf{0.065}_{(0.002)}$ | $\mathbf{0.065}_{(0.002)}$ |
| Concrete | $3.410_{(0.182)}$ | $1.927_{(0.086)}$ | $\mathbf{1.788}_{(0.077)}$ | $1.849_{(0.098)}$ | $\mathbf{1.741}_{(0.082)}$ |
| Energy | $0.519_{(0.043)}$ | $\mathbf{0.147}_{(0.006)}$ | $0.196_{(0.009)}$ | $\mathbf{0.143}_{(0.009)}$ | $0.157_{(0.008)}$ |
| Facebook | $4.022_{(0.099)}$ | $3.554_{(0.095)}$ | $3.211_{(0.059)}$ | $3.073_{(0.066)}$ | $\mathbf{2.977}_{(0.070)}$ |
| Kin8nm | $0.095_{(0.001)}$ | $0.061_{(0.001)}$ | $0.057_{(0.001)}$ | $\mathbf{0.051}_{(0.001)}$ | $\mathbf{0.051}_{(0.001)}$ |
| Life | $2.897_{(1.465)}$ | $0.815_{(0.027)}$ | $0.772_{(0.024)}$ | $0.794_{(0.023)}$ | $\mathbf{0.731}_{(0.022)}$ |
| MEPS | $\mathbf{5.527}_{(0.196)}$ | $6.448_{(0.092)}$ | $6.050_{(0.109)}$ | $6.150_{(0.114)}$ | $6.016_{(0.113)}$ |
| MSD | $4.524_{(0.005)}$ | $4.576_{(0.005)}$ | $4.363_{(0.004)}$ | $4.410_{(0.005)}$ | $\mathbf{4.347}_{(0.004)}$ |
| Naval | $0.003_{(0.000)}$ | $0.000_{(0.000)}$ | $0.000_{(0.000)}$ | $0.000_{(0.000)}$ | $\mathbf{0.000}_{(0.000)}$ |
| News | $\mathbf{2191}_{(47.5)}$ | $2361_{(52.6)}$ | $2346_{(52.6)}$ | $2545_{(41.0)}$ | $2380_{(52.1)}$ |
| Obesity | $3.208_{(0.028)}$ | $1.860_{(0.022)}$ | $\mathbf{1.740}_{(0.017)}$ | $1.866_{(0.021)}$ | $1.771_{(0.019)}$ |
| Power | $2.105_{(0.023)}$ | $1.531_{(0.019)}$ | $\mathbf{1.473}_{(0.022)}$ | $1.542_{(0.020)}$ | $\mathbf{1.471}_{(0.021)}$ |
| Protein | $5427_{(5409)}$ | $1.823_{(0.011)}$ | $1.788_{(0.009)}$ | $1.784_{(0.008)}$ | $\mathbf{1.742}_{(0.009)}$ |
| STAR | $132_{(1.589)}$ | $\mathbf{131}_{(1.380)}$ | $\mathbf{130}_{(1.283)}$ | $\mathbf{130}_{(1.214)}$ | $\mathbf{129}_{(1.198)}$ |
| Superconductor | $2.405_{(0.028)}$ | $\mathbf{0.126}_{(0.004)}$ | $0.150_{(0.004)}$ | $0.153_{(0.006)}$ | $\mathbf{0.128}_{(0.004)}$ |
| Synthetic | $5.779_{(0.042)}$ | $\mathbf{5.737}_{(0.039)}$ | $5.739_{(0.040)}$ | $\mathbf{5.731}_{(0.040)}$ | $5.730_{(0.040)}$ |
| Wave | $571020_{(883)}$ | $3891_{(73.9)}$ | $2349_{(10.3)}$ | $2679_{(16.0)}$ | $\mathbf{2026}_{(9.538)}$ |
| Wine | $0.385_{(0.005)}$ | $\mathbf{0.323}_{(0.005)}$ | $0.337_{(0.006)}$ | $0.322_{(0.006)}$ | $\mathbf{0.321}_{(0.006)}$ |
| Yacht | $1.177_{(0.158)}$ | $\mathbf{0.292}_{(0.042)}$ | $\mathbf{0.281}_{(0.048)}$ | $\mathbf{0.276}_{(0.048)}$ | $\mathbf{0.255}_{(0.046)}$ |
| IBUG W-T-L | 17-3-2 | 11-9-2 | 9-5-8 | - | 1-6-15 |
| IBUG+CBU W-T-L | 17-3-2 | 15-6-1 | 18-2-2 | 15-6-1 | - |

## 5.2 Probabilistic and Point Performance

We first compare IBUG's probabilistic and point predictions to each baseline on each dataset. See Table 1 for detailed CRPS results; due to space constraints, results for additional probabilistic metrics (e.g., NLL) as well as point performance results are in §B.3. Our main findings are as follows:

- On probabilistic performance, IBUG performs equally well or better than NGBoost and PGBM, winning on 17 and 11 (out of 22) datasets respectively, while losing on only 2 and 2 (respectively). Since CBU and IBUG performance is similar, we combine the two approaches, averaging their outputs; we denote this simple ensemble *IBUG+CBU*. Surprisingly, IBUG+CBU works very well, losing on only a maximum of 2 datasets when faced head-to-head against any other method; these results suggest IBUG and CBU are complimentary approaches.

- On point performance, PGBM, CBU, and IBUG performed significantly better than NGBoost; this is consistent with previous work and is perhaps unsurprising since NGBoost is optimized for probabilistic performance, not point performance. However, IBUG generally performed better than PGBM, winning on 13 datasets and losing on only 1 dataset; and performed slightly better than CBU, winning on 6 datasets with no losses.

We also compare IBUG with two additional baselines—$k$NN and BART [13]—shown in §C.2–C.3 due to space constraints. We find IBUG generally outperforms these methods in both probabilistic and point performance. Overall, the results in this section suggest IBUG generates both competitive probabilistic and point predictions compared to existing methods.

Table 2: Probabilistic (CRPS, NLL) performance on the test set for IBUG using different base models. Results are averaged over 10 folds, and standard errors are shown in subscripted parentheses; lower is better. On 6 and 5 datasets, respectively, either IBUG-LightGBM or IBUG-XGBoost significantly outperforms IBUG-CatBoost on the validation set and subsequently on the test set, demonstrating the potential for improved probabilistic performance by using IBUG with different base models.

| | Test CRPS ($\downarrow$) | | | | Test NLL ($\downarrow$) | | |
|---|---|---|---|---|---|---|---|
| Dataset | CatBoost | LightGBM | XGBoost | Dataset | CatBoost | LightGBM | XGBoost |
| Bike | $0.974_{(0.048)}$ | $\mathbf{0.819}_{(0.024)}$ | $\mathbf{0.849}_{(0.012)}$ | Bike | $1.886_{(0.056)}$ | $\mathbf{1.292}_{(0.048)}$ | $\mathbf{1.662}_{(0.024)}$ |
| MSD | $4.410_{(0.005)}$ | $\mathbf{4.372}_{(0.005)}$ | $4.418_{(0.005)}$ | MSD | $3.415_{(0.002)}$ | $\mathbf{3.409}_{(0.002)}$ | $\mathbf{3.402}_{(0.002)}$ |
| News | $2545_{(41.0)}$ | $\mathbf{2436}_{(50.8)}$ | $2551_{(56.0)}$ | Naval | $-6.208_{(0.010)}$ | $\mathbf{-6.281}_{(0.007)}$ | $-5.853_{(0.014)}$ |
| Power | $1.542_{(0.020)}$ | $1.536_{(0.022)}$ | $\mathbf{1.518}_{(0.018)}$ | Obesity | $2.646_{(0.009)}$ | $\mathbf{2.593}_{(0.016)}$ | $\mathbf{2.624}_{(0.010)}$ |
| Protein | $1.784_{(0.008)}$ | $\mathbf{1.683}_{(0.009)}$ | $1.788_{(0.008)}$ | Supercon. | $0.783_{(0.181)}$ | $\mathbf{-0.496}_{(0.169)}$ | $20.4_{(23.2)}$ |
| Supercon. | $0.153_{(0.006)}$ | $\mathbf{0.090}_{(0.005)}$ | $\mathbf{0.010}_{(0.003)}$ | | | | |

Figure 1: *Left*: Distribution of the $k$-nearest training examples for 5 randomly-selected test instances from the MEPS (top) and Wine (bottom) datasets. *Right*: Test NLL (with standard error) when modeling the posterior using two different distributions (lower is better). IBUG can model parametric *and* non-parametric distributions that better fit the underlying data than assuming normality.

## 5.3 Different Base Models

Here we experiment using different base models for IBUG besides CatBoost [52]; specifically, we use LightGBM [36] and XGBoost [12], two popular gradient boosting frameworks. Table 2 shows that using a different base model can result in improved probabilistic performance. This highlights IBUG's agnosticism to GBRT type, enabling practitioners to apply IBUG to future models with improved point prediction performance.

## 5.4 Posterior Modeling

One of the unique benefits of IBUG is the ability to directly model the output using empirical samples (Figure 1), giving practitioners a better sense of the output distribution for specific predictions. IBUG can optimize a distribution *after* training, and has more flexibility in the types of distributions it can model—from distributions using just location and scale to those with high-order moments as well as non-parametric density estimators. To test this flexibility, we model each probabilistic prediction using the following distributions: normal, skewnormal, lognormal, Laplace, student t, logistic, Gumbel, Weibull, and KDE; we then select the distribution with the best average NLL on the validation set, and evaluate its probabilistic performance on the test set.

Figure 1 demonstrates that the selected distributions for the MEPS and Wine datasets achieve better probabilistic performance than assuming normality. Qualitatively, the empirical densities of $A^{(k)}$ for a randomly sampled set of test instances reaffirms the selected distributions. As an additional comparison, we report CBU achieves a test NLL of $3.699_{\pm 0.038}$ and $1.025_{\pm 0.028}$ for the MEPS and wine datasets (respectively) using a normal distribution, while IBUG achieves $-6.887_{\pm 0.260}$ and $0.785_{\pm 0.025}$ using Weibull and KDE estimation (respectively). For the MEPS dataset, the selected Weibull distribution takes a shape parameter, which IBUG estimates directly on a *per prediction* basis using $A^{(k)}$ and MLE. In contrast, PGBM or CBU would need to optimize a global shape value using a validation set, which is likely to be suboptimal for individual predictions.

Table 3: Probabilistic performance comparison of each method with vs. without variance calibration. In all cases, calibration maintains or improves performance; it is especially helpful for CBU.

|  | CRPS | | | NLL | | |
|---|---|---|---|---|---|---|
| Method | Wins | ties | Losses | Wins | Ties | Losses |
| NGBoost | 9 | 13 | 0 | 1 | 21 | 0 |
| PGBM | 13 | 9 | 0 | 11 | 11 | 0 |
| CBU | 17 | 5 | 0 | 11 | 11 | 0 |
| IBUG | 13 | 9 | 0 | 5 | 17 | 0 |

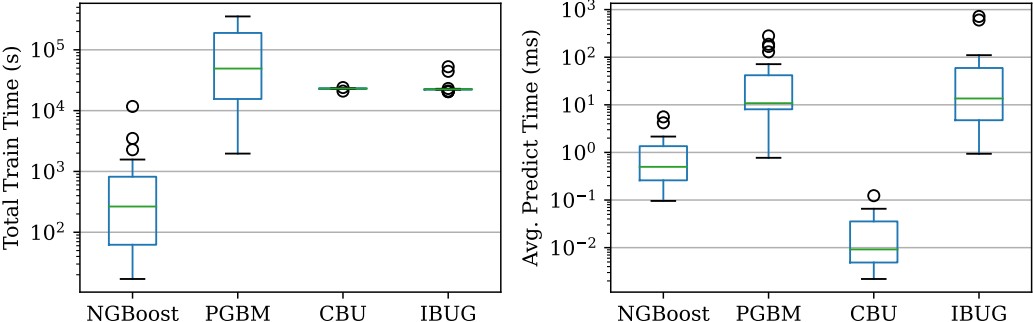

Figure 2: Runtime comparison. *Left*: Total train time (including tuning). *Right*: Average prediction time per test example. Results are shown for all datasets, averaged over 10 folds (exact values are in §B.4, Tables 12 and 13). On average, IBUG has comparable training times to PGBM and CBU, but is relatively slow for prediction.

## 5.5 Variance Calibration

Table 3 shows probabilistic performance comparisons of each method against itself with and without variance calibration. In all cases, variance calibration (§3.2) either maintains or improves performance for all methods, especially CBU. Overall, these results suggest that variance calibration should be a standard procedure for probabilistic prediction, unless using a method that has particularly well-calibrated predictions to begin with. We therefore use variance calibration in all of our results.

Additionally, §C.1 shows performance results for all methods *without* variance calibration. Overall, we observe similar relative performance trends as when applying calibration (Table 1).

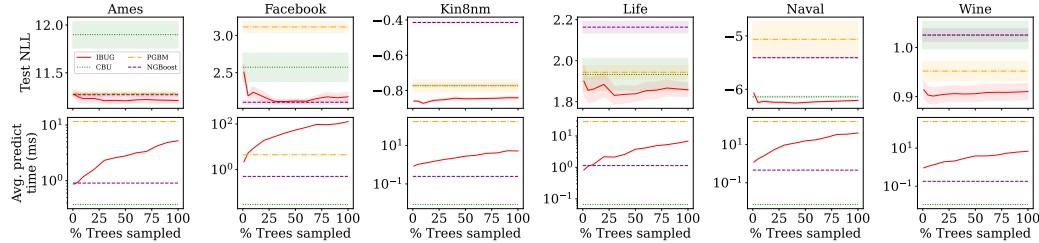

Figure 3: Change in probabilistic (NLL) performance (top) and average prediction time (in seconds) per test example (bottom) as a function of $\tau$ for six datasets with trees sampled *first-to-last*; lower is better. NGBoost, PGBM, and CBU are added for additional context. The shaded regions represent the standard error. Overall, average prediction time decreases significantly as $\tau$ decreases while test NLL often remains relatively stable, enabling IBUG to generate probabilistic predictions with significant increased efficiency.

### 5.6 Sampling Trees

Figure 2 shows the runtime for each method broken down into total training time (including tuning) and prediction time per test example. On average, IBUG has similar training times to PGBM and CBU, but on some datasets, IBUG is roughly an order of magnitude faster than PGBM. For predictions, IBUG is similar to PGBM but relatively slow compared to NGBoost and CBU.

However, by sampling $\tau < T$ trees when computing the affinity vector, IBUG can significantly reduce prediction time. Figure 3 shows results when sampling trees first-to-last, which typically works best over all tree-sampling strategies (alternate sampling strategies are evaluated in §C.4). As $\tau$ decreases, we observe average prediction time decreases roughly 1-2 orders of magnitude while probabilistic performance remains relatively stable until $\tau/T$ reaches roughly 1–5%, at which point probabilistic performance sometimes starts to decrease more rapidly. Note for the Ames and Life datasets, IBUG can reach the same average prediction time as NGBoost while maintaining the same or better probabilistic performance than NGBoost, PGBM, and CBU. These results demonstrate that if speed is a concern, IBUG can approximate the affinity computation to speed up prediction times while maintaining competitive probabilistic performance.

## 6 Additonal Related Work

Traditional approaches to probabilistic regression include generalized additive models for location, scale, and shape (GAMLSS), which allow for a flexible choice of distribution for the target variable but are restricted to pre-specified model forms [56]. Prophet [68] also produces probabilistic estimates for generalized additive models, but has been shown to underperform as compared to more recent approaches [60, 3]. Bayesian methods [48, 30] naturally generate uncertainty estimates by integrating over the posterior; however, exact solutions are limited to simple models, and more complex models such as Bayesian Additive Regression Trees (BART) [13, 41] require computationally expensive sampling techniques (e.g., MCMC [4]) to provide approximate solutions.

Other approaches to probabilistic regression tasks include conformal predictions [61, 67, 5] which produce confidence intervals via empirical errors obtained in the past, and quantile regression [32, 37, 45, 57]. Similar to PGBM, distributional forests (DFs) [59] estimate distributional parameters in each leaf, and average these estimates over all trees in the forest. Deep learning approaches for probabilistic regression [54, 74, 3] have increased recently, with notable approaches such as DeepAR [58] and methods based on transformer architectures [38, 39].

## 7 Conclusion

IBUG uses ideas from instance-based learning to enable probabilistic predictions for *any* GBRT point predictor. IBUG generates probabilistic predictions by using the $k$-nearest training instances to the test instance found using the structure of the trees in the ensemble. Our results on 22 regression datasets demonstrate this simple wrapper produces competitive probabilistic and point predictions to current state-of-the-art methods, most notably NGBoost [22], PGBM [64], and CBU [44]. We also show that IBUG can more flexibly model the posterior distribution of a prediction using any parametric *or* non-parametric density estimator. IBUG's one limitation is relatively slow prediction time. However, we show that approximations in the search for the $k$-nearest training instances can significantly speed up prediction time; predictions are also easily parallelizable in IBUG. For future work, we plan to investigate other approximations to the affinity vector such as subsampling or even reweighting training instances, which may lead to significant speed ups of IBUG.

## Acknowledgments and Disclosure of Funding

We would like to thank Zayd Hammoudeh for useful discussions and feedback and the reviewers for their constructive comments that improved this paper. This work was supported by a grant from the Air Force Research Laboratory and the Defense Advanced Research Projects Agency (DARPA)—agreement number FA8750-16-C-0166, subcontract K001892-00-S05, as well as a second grant from DARPA, agreement number HR00112090135. This work benefited from access to the University of Oregon high-performance computer, Talapas.

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
