# A   Algorithmic Details

Figure 4 summarizes how IBUG generates a probabilistic prediction for a given input instance.

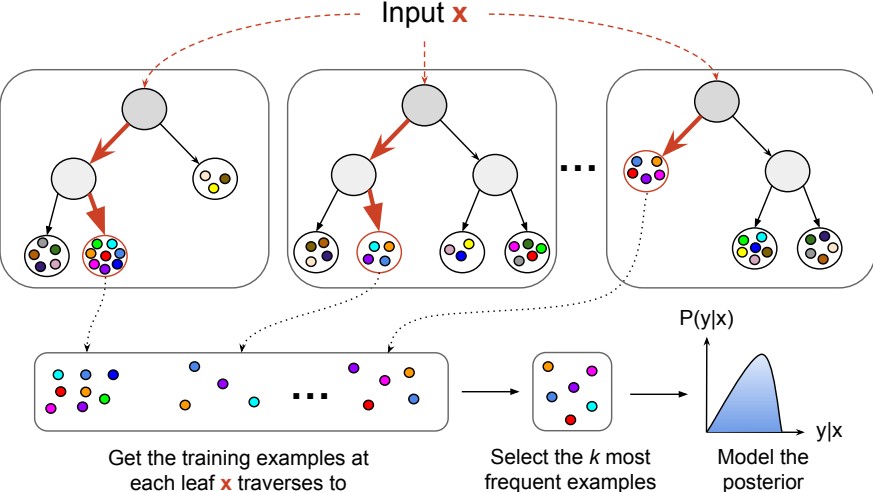

Figure 4: IBUG workflow. Given a GBRT model and an input instance $x$, IBUG collects the training examples at each leaf $x$ traverses to, keeps the $k$ most frequent examples, and then uses those examples to model the output distribution.

## A.1   Ethical Statement

In general, this work has no foreseeable negative societal impacts; however, users should carefully validate their models as imprecise uncertainty estimates may adversely affect certain domains (e.g., healthcare, weather).

# B  Implementation and Experiment Details

We implement IBUG in Python, using Cython—a Python package allowing the development of C extensions—to store a unified representation of the model structure. IBUG currently supports all major modern gradient boosting frameworks including XGBoost [12], LightGBM [36], and CatBoost [52]. Experiments are run on an Intel(R) Xeon(R) CPU E5-2690 v4 @ 2.6GHz with 60GB of RAM @ 2.4GHz. We run our experiments on publicly available datasets. Links to all data sources as well as the code for IBUG and all experiments is currently available online at `https://github.com/jjbrophy47/ibug`.

**Metrics.**  We use the continuous ranked probability score (CRPS) and negative log likelihood (NLL) to measure probabilistic performance. CRPS is a quadratic measure of discrepancy between the cumulative distribution function (CDF) $F$ of forecast $\hat{y}$ and the empirical CDF of the scalar observation $y$: $\int (F(\hat{y}) - \mathbb{1}[\hat{y} \geq y])^2 d\hat{y}$ in which $\mathbb{1}$ is the indicator function [29, 76]. To evaluate point performance, we use root mean squared error (RMSE): $\sqrt{\frac{1}{n} \sum_{i=1}^{n} (y_i - \hat{y}_i)^2}$.

## B.1  Datasets

This section gives a detailed description for each dataset we use in our experiments.

- **Ames** [19] consists of 2,930 instances of housing prices in the Ames, Iowa area characterized by 80 attributes. The aim is to predict the sale price of a given house.

- **Bike** [23, 21] contains 17,379 measurements of the number of bikes rented per hour characterized by 16 attributes. The aim is to predict the number of bikes rented for a given hour.

- **California** [49] consists of 20,640 instances of median housing prices in various California districts characterized by 8 attributes. The aim is to predict the median housing price for the given district.

- **Communities** [55, 21] consists of 1,994 measurements of violent crime statistics based on crime, survey, and census data. The dataset is characterized by 100 attributes, and the aim is to predict the violent crime rate for a given population.

- **Concrete** [75, 21] consists of 1,030 instances of concrete characterized by 8 attributes. The aim is to predict the compressive strength of the concrete.

- **Energy** [70, 21] consists of 768 buildings in which each building is one of 12 different shapes and is characterized by 8 features. The aim is to predict the cooling load associated with the building.

- **Facebook** [63, 21] consists of 40,949 Facebook posts characterized by 53 attributes. The aim is to predict the number of comments for a given post.

- **Kin8nm** [73] consists of 8,192 instances of the forward kinematics of an 8 link robotic arm. The aim is to predict the forward kinematics of the robotic arm.

- **Life** [53] consists of 2,928 instances of life expectancy estimates for various countries during one year. Each instance is characterized by 20 attributes, and the aim is to predict the life expectancy of the country during a specific year.

- **MEPS** [15] consists of 16,656 instances of medical expenditure survey data. Each instance is characterized by 139 attributes, and the aim is to predict the insurance utilization for the given medical expenditure.

- **MSD** [7] consists of 515,345 songs characterized by 90 audio features constructed from each song. The aim is to predict what year the song was released based on the audio features.

- **Naval** [16, 21] consists of 11,934 instances extracted from a high-performing gas turbine simulation. Each instance is characterized by 16 features. The aim is to predict the gas turbine decay coefficient.

- **News** [21, 24] consists of 39,644 Mashable articles characterized by 60 features. The aim is to predict the number of shares for a given article.

- **Obesity** [66] contains 48,346 instances of obesity rates for different states and regions with differing socioeconomic backgrounds. Each instance is characterized by 32 attributes. The aim is to predict the obesity rate of the region.

- **Power** [21, 35, 71] contains 9,568 readings of a Combined Cycle Power Plant (CCPP) at full work load. Each reading is characterized by 4 features. The aim is to predict the net hourly electrical energy output.

- **Protein** [21] contains 45,730 tertiary-protein-structure instances characterized by 9 attributes. The aim is to predict the armstrong coefficient of the protein structure.

- **STAR** [21, 65] contains 2,161 student-teacher achievement scores characterized by 39 attributes. The aim is to predict the student-teacher achievement based on the given intervention.

- **Superconductor** [21, 31] contains 21,263 potential superconductors characterized by 81 attributes. The aim is to predict the critical temperature of the given superconductor.

- **Synthetic** [10, 27] is a non-linear synthetic regression dataset in which the inputs are independent and uniformly distributed on the interval $[0, 1]$; the dataset contains 10,000 instances characterized by 100 attributes.

- **Wave** [21] consists of 287,999 positions and absorbed power outputs of wave energy converters (WECs) in four real wave scenarios off the southern coast of Australia (Sydney, Adelaide, Perth and Tasmania). The aim is to predict the total power output of a given WEC.

- **Wine** [17, 21] consists of 6,497 instances of Portuguese "Vinho Verde" red and white wine characterized by 11 features. The aim is to predict the quality of the wine from 0-10.

- **Yacht** [21] consists of 308 instances of yacht-sailing performance characterized by 6 attributes. The aim is to predict the residual resistance per unit weight of displacement.

For each dataset, we generate one-hot encodings for any categorical variable and leave all numeric and binary variables as is. Table 4 shows a summary of the datasets after preprocessing.

Table 4: Dataset summary after preprocessing.

| Dataset | Source | $n$ | $p$ |
|---|---|---|---|
| Ames | [19] | 2,930 | 358 |
| Bike | [23, 21] | 17,379 | 37 |
| California | [49] | 20,640 | 100 |
| Communities | [21, 55] | 1,994 | 100 |
| Concrete | [75, 21] | 1,030 | 8 |
| Energy | [70, 21] | 768 | 16 |
| Facebook | [63, 21] | 40,949 | 133 |
| Kin8nm | [73] | 8,192 | 8 |
| Life | [53] | 2,928 | 204 |
| MEPS | [15] | 15,656 | 139 |
| MSD | [7] | 515,345 | 90 |
| Naval | [21, 16] | 11,934 | 17 |
| News | [21, 24] | 39,644 | 58 |
| Obesity | [66] | 48,346 | 100 |
| Power | [21, 35, 71] | 9,568 | 4 |
| Protein | [21] | 45,730 | 9 |
| STAR | [21, 65] | 2,161 | 95 |
| Superconductor | [21, 31] | 21,263 | 82 |
| Synthetic | [10, 27] | 10,000 | 100 |
| Wave | [21] | 287,999 | 48 |
| Wine | [17, 21] | 6,497 | 11 |
| Yacht | [21] | 308 | 6 |

## B.2 Hyperparameters

Tables 5 and 6 show hyperparameter values selected most often for each dataset when optimizing CRPS and NLL, respectively. We tune nearest-neighbor hyperparameter $k$ using values [3, 5, 7, 9, 11, 15, 31, 61, 91, 121, 151, 201, 301, 401, 501, 601, 701], $\gamma$ and $\delta$ using values [1e-8, 1e-7, 1e-6, 1e-5, 1e-4, 1e-3, 1e-2, 1e-1, 0, 1e0, 1e1, 1e2, 1e3] with multipliers [1.0, 2.5, 5.0], number of trees $T$ using values [10, 25, 50, 100, 250, 500, 1000, 2000] (since NGBoost has no hyperparameters to tune besides $T$, we tune $T$ on the validation set using early stopping [22]), learning rate $\eta$ using values [0.01, 0.1], maximum number of leaves $h$ using values [15, 31, 61, 91], minimum number of leaves $n_{\ell_0}$ using values [1, 20], maximum depth $d$ using values [2, 3, 5, 7, -1 (unlimited)], and $\rho$ which selects the minimum variance computed from the validation set predictions. For the MSD and Wave datasets, we use a bagging fraction of 0.1 [22, 64].

Table 5: Hyperparameters selected most often over 10 folds for each dataset when optimizing CRPS.

| | NGBoost | | PGBM | | | | | CBU | | | | |
|---|---|---|---|---|---|---|---|---|---|---|---|---|
| Dataset | $T$ | $\gamma/\delta$ | $T$ | $\eta$ | $h$ | $n_{\ell_0}$ | $\gamma/\delta$ | $T$ | $\eta$ | $d$ | $n_{\ell_0}$ | $\gamma/\delta$ |
| Ames | 2000 | $\gamma$:1e+00 | 2000 | 0.1 | 15 | 1 | $\gamma$:2e+00 | 2000 | 0.1 | 7 | 1 | $\delta$:5e+03 |
| Bike | 2000 | $\gamma$:5e-01 | 2000 | 0.01 | 61 | 1 | $\delta$:1e-08 | 2000 | 0.1 | 2 | 1 | $\delta$:5e-02 |
| California | 2000 | $\delta$:3e-02 | 1000 | 0.1 | 31 | 20 | $\delta$:3e-02 | 2000 | 0.1 | -1 | 1 | $\delta$:1e-01 |
| Communities | 223 | $\delta$:1e-02 | 500 | 0.01 | 15 | 20 | $\gamma$:1e+01 | 2000 | 0.01 | 7 | 1 | $\delta$:5e-02 |
| Concrete | 2000 | $\delta$:1e+00 | 2000 | 0.1 | 15 | 20 | $\delta$:1e-08 | 2000 | 0.1 | 5 | 1 | $\delta$:2e+00 |
| Energy | 2000 | $\gamma$:5e-01 | 2000 | 0.1 | 15 | 1 | $\gamma$:5e-01 | 2000 | 0.1 | 3 | 1 | $\delta$:1e-01 |
| Facebook | 2000 | $\gamma$:1e+00 | 2000 | 0.01 | 15 | 1 | $\gamma$:2e+00 | 2000 | 0.1 | 5 | 1 | $\gamma$:1e+00 |
| Kin8nm | 581 | $\delta$:3e-02 | 2000 | 0.1 | 61 | 20 | $\delta$:5e-02 | 2000 | 0.1 | 7 | 1 | $\delta$:5e-02 |
| Life | 2000 | $\gamma$:1e+00 | 2000 | 0.1 | 15 | 1 | $\delta$:2e-01 | 2000 | 0.1 | 5 | 1 | $\delta$:1e+00 |
| MEPS | 583 | $\delta$:1e-01 | 50 | 0.1 | 15 | 1 | $\delta$:1e-08 | 100 | 0.01 | -1 | 1 | $\gamma$:1e+00 |
| MSD | 2000 | $\delta$:1e-01 | 2000 | 0.01 | 91 | 20 | $\gamma$:1e+01 | 2000 | 0.1 | 7 | 1 | $\delta$:5e-01 |
| Naval | 2000 | $\delta$:0e+00 | 2000 | 0.1 | 61 | 20 | $\gamma$:3e-02 | 2000 | 0.1 | 7 | 1 | $\delta$:3e-04 |
| News | 2000 | $\gamma$:5e-01 | 100 | 0.01 | 15 | 20 | $\delta$:2e+03 | 100 | 0.01 | 2 | 1 | $\gamma$:5e-01 |
| Obesity | 2000 | $\delta$:1e-01 | 500 | 0.1 | 91 | 20 | $\delta$:1e-08 | 2000 | 0.1 | 7 | 1 | $\delta$:5e-01 |
| Power | 2000 | $\delta$:2e-01 | 500 | 0.1 | 91 | 1 | $\delta$:5e-01 | 2000 | 0.1 | 7 | 1 | $\delta$:1e+00 |
| Protein | 2000 | $\delta$:1e-01 | 2000 | 0.1 | 91 | 20 | $\gamma$:2e+00 | 2000 | 0.1 | 7 | 1 | $\delta$:1e+00 |
| STAR | 187 | $\delta$:2e+01 | 1000 | 0.01 | 15 | 1 | $\gamma$:1e+01 | 2000 | 0.01 | -1 | 1 | $\delta$:5e+01 |
| Superconductor | 162 | $\gamma$:5e-01 | 1000 | 0.01 | 15 | 20 | $\delta$:5e-02 | 2000 | 0.1 | -1 | 1 | $\delta$:3e-02 |
| Synthetic | 208 | $\delta$:5e-01 | 500 | 0.01 | 15 | 20 | $\delta$:1e+01 | 2000 | 0.01 | 3 | 1 | $\delta$:1e+00 |
| Wave | 2000 | $\delta$:5e+03 | 2000 | 0.1 | 15 | 1 | $\delta$:1e-08 | 2000 | 0.1 | -1 | 1 | $\delta$:2e+02 |
| Wine | 309 | $\delta$:5e-02 | 2000 | 0.01 | 91 | 20 | $\delta$:5e-01 | 2000 | 0.1 | 7 | 1 | $\delta$:2e-01 |
| Yacht | 2000 | $\gamma$:5e-01 | 2000 | 0.1 | 15 | 1 | $\delta$:0e+00 | 2000 | 0.1 | 3 | 1 | $\gamma$:2e+00 |

| | CatBoost | | | | IBUG | | |
|---|---|---|---|---|---|---|---|
| Dataset | $T$ | $\eta$ | $d$ | $n_{\ell_0}$ | $k$ | $\rho$ | $\gamma/\delta$ |
| Ames | 2000 | 0.1 | -1 | 1 | 5 | 2206 | $\delta$:3e-04 |
| Bike | 2000 | 0.1 | 2 | 1 | 3 | 0.471 | $\gamma$:2e-01 |
| California | 2000 | 0.1 | -1 | 1 | 7 | 2e-15 | $\delta$:1e-08 |
| Communities | 2000 | 0.01 | -1 | 1 | 15 | 0.017 | $\delta$:0e+00 |
| Concrete | 2000 | 0.1 | 5 | 1 | 3 | 0.049 | $\gamma$:5e-01 |
| Energy | 2000 | 0.1 | 5 | 1 | 3 | 0.035 | $\gamma$:1e-01 |
| Facebook | 2000 | 0.1 | -1 | 1 | 15 | 0.213 | $\delta$:1e-01 |
| Kin8nm | 2000 | 0.1 | 7 | 1 | 3 | 0.003 | $\gamma$:5e-01 |
| Life | 2000 | 0.1 | 5 | 1 | 3 | 0.047 | $\gamma$:5e-01 |
| MEPS | 250 | 0.01 | 5 | 1 | 201 | 1.08 | $\delta$:1e-07 |
| MSD | 2000 | 0.1 | 7 | 1 | 31 | 1.25 | $\delta$:1e-07 |
| Naval | 2000 | 0.1 | 7 | 1 | 3 | 1e-15 | $\gamma$:5e-01 |
| News | 1000 | 0.01 | 2 | 1 | 15 | 163 | $\gamma$:5e-01 |
| Obesity | 2000 | 0.1 | 7 | 1 | 5 | 0.306 | $\gamma$:5e-01 |
| Power | 2000 | 0.1 | 7 | 1 | 5 | 0.220 | $\delta$:1e-01 |
| Protein | 2000 | 0.1 | 7 | 1 | 31 | 0.028 | $\delta$:1e-01 |
| STAR | 250 | 0.01 | 5 | 1 | 121 | 192 | $\delta$:1e-08 |
| Superconductor | 2000 | 0.1 | 5 | 1 | 3 | 5e-15 | $\gamma$:1e-01 |
| Synthetic | 1000 | 0.01 | 7 | 1 | 401 | 9.34 | $\delta$:1e-08 |
| Wave | 2000 | 0.1 | -1 | 1 | 3 | 2e-10 | $\gamma$:2e-01 |
| Wine | 2000 | 0.1 | 7 | 1 | 15 | 0.268 | $\delta$:2e-08 |
| Yacht | 2000 | 0.1 | 2 | 1 | 3 | 0.196 | $\gamma$:1e-01 |

Table 6: Hyperparameters selected most often over 10 folds for each dataset when optimizing NLL.

| | NGBoost | | PGBM | | | | | CBU | | | | |
|---|---|---|---|---|---|---|---|---|---|---|---|---|
| Dataset | $T$ | $\gamma/\delta$ | $T$ | $\eta$ | $h$ | $n_{\ell_0}$ | $\gamma/\delta$ | $T$ | $\eta$ | $d$ | $n_{\ell_0}$ | $\gamma/\delta$ |
| Ames | 373 | $\delta$:2e+03 | 2000 | 0.1 | 15 | 1 | $\gamma$:2e+00 | 2000 | 0.1 | 7 | 1 | $\gamma$:1e+01 |
| Bike | 926 | $\delta$:0e+00 | 2000 | 0.01 | 61 | 1 | $\delta$:1e-08 | 2000 | 0.1 | 2 | 1 | $\delta$:0e+00 |
| California | 2000 | $\delta$:5e-02 | 1000 | 0.1 | 31 | 20 | $\delta$:1e-01 | 2000 | 0.1 | -1 | 1 | $\delta$:2e-01 |
| Communities | 156 | $\delta$:1e-02 | 500 | 0.01 | 15 | 20 | $\gamma$:1e+01 | 2000 | 0.01 | 7 | 1 | $\delta$:1e-01 |
| Concrete | 383 | $\delta$:1e+00 | 2000 | 0.1 | 15 | 20 | $\delta$:1e+00 | 2000 | 0.1 | 5 | 1 | $\delta$:2e+00 |
| Energy | 422 | $\delta$:1e-02 | 2000 | 0.1 | 15 | 1 | $\delta$:1e-08 | 2000 | 0.1 | 3 | 1 | $\delta$:1e-01 |
| Facebook | 549 | $\delta$:0e+00 | 2000 | 0.01 | 15 | 1 | $\gamma$:5e+00 | 2000 | 0.1 | 5 | 1 | $\gamma$:2e+00 |
| Kin8nm | 975 | $\delta$:1e-02 | 2000 | 0.1 | 61 | 20 | $\delta$:5e-02 | 2000 | 0.1 | 7 | 1 | $\delta$:1e-01 |
| Life | 366 | $\delta$:2e-01 | 2000 | 0.1 | 15 | 1 | $\delta$:1e+00 | 2000 | 0.1 | 5 | 1 | $\delta$:1e+00 |
| MEPS | 188 | $\delta$:1e+00 | 50 | 0.1 | 15 | 1 | $\delta$:1e-08 | 100 | 0.01 | -1 | 1 | $\delta$:1e+00 |
| MSD | 2000 | $\delta$:3e-02 | 2000 | 0.01 | 91 | 20 | $\gamma$:1e+01 | 2000 | 0.1 | 7 | 1 | $\delta$:1e+00 |
| Naval | 2000 | $\delta$:5e-05 | 2000 | 0.1 | 61 | 20 | $\gamma$:5e-02 | 2000 | 0.1 | 7 | 1 | $\delta$:3e-04 |
| News | 38 | $\delta$:1e+03 | 100 | 0.01 | 15 | 20 | $\gamma$:1e+01 | 100 | 0.01 | 2 | 1 | $\delta$:2e+03 |
| Obesity | 2000 | $\delta$:0e+00 | 500 | 0.1 | 91 | 20 | $\delta$:1e-08 | 2000 | 0.1 | 7 | 1 | $\delta$:5e-01 |
| Power | 275 | $\delta$:2e-01 | 500 | 0.1 | 91 | 1 | $\delta$:1e+00 | 2000 | 0.1 | 7 | 1 | $\delta$:2e+00 |
| Protein | 2000 | $\delta$:2e-01 | 2000 | 0.1 | 91 | 20 | $\delta$:2e+00 | 2000 | 0.1 | 7 | 1 | $\delta$:1e+00 |
| STAR | 176 | $\delta$:1e+01 | 1000 | 0.01 | 15 | 1 | $\delta$:2e+02 | 2000 | 0.01 | -1 | 1 | $\delta$:5e+01 |
| Superconductor | 378 | $\gamma$:1e+00 | 1000 | 0.01 | 15 | 20 | $\gamma$:2e+00 | 2000 | 0.1 | -1 | 1 | $\delta$:1e-01 |
| Synthetic | 284 | $\delta$:5e-01 | 500 | 0.01 | 15 | 20 | $\delta$:1e+01 | 2000 | 0.01 | 3 | 1 | $\delta$:1e+00 |
| Wave | 2000 | $\gamma$:1e+00 | 2000 | 0.1 | 15 | 1 | $\delta$:5e+02 | 2000 | 0.1 | -1 | 1 | $\delta$:2e+02 |
| Wine | 390 | $\delta$:5e-02 | 2000 | 0.01 | 91 | 20 | $\gamma$:2e+01 | 2000 | 0.1 | 7 | 1 | $\delta$:5e-01 |
| Yacht | 356 | $\delta$:0e+00 | 2000 | 0.1 | 15 | 1 | $\delta$:5e-02 | 2000 | 0.1 | 3 | 1 | $\delta$:5e-01 |

| | CatBoost | | | | IBUG | | |
|---|---|---|---|---|---|---|---|
| Dataset | $T$ | $\eta$ | $d$ | $n_{\ell_0}$ | $k$ | $\rho$ | $\gamma/\delta$ |
| Ames | 2000 | 0.1 | -1 | 1 | 11 | 4673 | $\delta$:1e-08 |
| Bike | 2000 | 0.1 | 2 | 1 | 5 | 0.4 | $\gamma$:2e-01 |
| California | 2000 | 0.1 | -1 | 1 | 31 | 0.063 | $\delta$:0e+00 |
| Communities | 2000 | 0.01 | -1 | 1 | 61 | 0.026 | $\delta$:0e+00 |
| Concrete | 2000 | 0.1 | 5 | 1 | 5 | 0.56 | $\delta$:1e-08 |
| Energy | 2000 | 0.1 | 5 | 1 | 3 | 0.087 | $\gamma$:2e-01 |
| Facebook | 2000 | 0.1 | -1 | 1 | 301 | 0.175 | $\delta$:1e-01 |
| Kin8nm | 2000 | 0.1 | 7 | 1 | 7 | 0.031 | $\delta$:0e+00 |
| Life | 2000 | 0.1 | 5 | 1 | 7 | 0.22 | $\delta$:2e-08 |
| MEPS | 250 | 0.01 | 5 | 1 | 301 | 1.76 | $\delta$:1e+00 |
| MSD | 2000 | 0.1 | 7 | 1 | 61 | 1.75 | $\delta$:1e-07 |
| Naval | 2000 | 0.1 | 7 | 1 | 5 | 4e-04 | $\gamma$:5e-01 |
| News | 1000 | 0.01 | 2 | 1 | 301 | 994 | $\delta$:2e+03 |
| Obesity | 2000 | 0.1 | 7 | 1 | 9 | 0.529 | $\delta$:1e-07 |
| Power | 2000 | 0.1 | 7 | 1 | 15 | 0.861 | $\delta$:1e-07 |
| Protein | 2000 | 0.1 | 7 | 1 | 121 | 0.218 | $\delta$:5e-08 |
| STAR | 250 | 0.01 | 5 | 1 | 121 | 189 | $\delta$:1e-05 |
| Superconductor | 2000 | 0.1 | 5 | 1 | 7 | 0.019 | $\gamma$:2e-01 |
| Synthetic | 1000 | 0.01 | 7 | 1 | 401 | 9.39 | $\delta$:1e-08 |
| Wave | 2000 | 0.1 | -1 | 1 | 31 | 349 | $\gamma$:2e-01 |
| Wine | 2000 | 0.1 | 7 | 1 | 61 | 0.297 | $\delta$:2e-08 |
| Yacht | 2000 | 0.1 | 2 | 1 | 3 | 0.196 | $\gamma$:2e-01 |

## B.3 Additional Metrics

In this section, we show results for point performance and probabilistic performance with additional metrics. Each table shows average results over the 10 random folds for each dataset, with standard errors in subscripted parentheses. We use the *Uncertainty Toolbox*[3] [14] to compute each metric. Lower is better for all metrics.

**Point performance and negative-log likelihood.** Tables 7 and 8 show point (RMSE) and probabilistic (NLL) performance of each method.

Table 7: Point (RMSE ↓) performance for each method on each dataset.

| Dataset | NGBoost | PGBM | CBU | IBUG | IBUG+CBU |
|---|---|---|---|---|---|
| Ames | 24580$_{(804)}$ | **23541**$_{(1225)}$ | **22576**$_{(924)}$ | **22942**$_{(1388)}$ | **22391**$_{(1119)}$ |
| Bike | 4.173$_{(0.076)}$ | 3.812$_{(0.225)}$ | **2.850**$_{(0.192)}$ | **2.826**$_{(0.200)}$ | **2.708**$_{(0.202)}$ |
| California | 0.503$_{(0.003)}$ | 0.445$_{(0.001)}$ | 0.449$_{(0.002)}$ | **0.432**$_{(0.001)}$ | 0.434$_{(0.002)}$ |
| Communities | 0.137$_{(0.004)}$ | **0.135**$_{(0.004)}$ | **0.133**$_{(0.004)}$ | **0.133**$_{(0.004)}$ | **0.132**$_{(0.004)}$ |
| Concrete | 5.485$_{(0.182)}$ | 3.840$_{(0.209)}$ | **3.682**$_{(0.202)}$ | **3.629**$_{(0.183)}$ | **3.617**$_{(0.188)}$ |
| Energy | 0.461$_{(0.030)}$ | 0.291$_{(0.022)}$ | 0.381$_{(0.023)}$ | **0.264**$_{(0.023)}$ | 0.303$_{(0.023)}$ |
| Facebook | **20.8**$_{(1.102)}$ | 20.5$_{(0.867)}$ | 20.1$_{(0.913)}$ | 20.0$_{(0.903)}$ | 19.9$_{(0.929)}$ |
| Kin8nm | 0.176$_{(0.001)}$ | 0.108$_{(0.001)}$ | 0.103$_{(0.001)}$ | **0.086**$_{(0.001)}$ | 0.091$_{(0.001)}$ |
| Life | 2.280$_{(0.032)}$ | 1.678$_{(0.059)}$ | **1.637**$_{(0.058)}$ | **1.652**$_{(0.055)}$ | **1.610**$_{(0.056)}$ |
| MEPS | **23.7**$_{(0.955)}$ | **24.1**$_{(0.760)}$ | 23.5$_{(0.950)}$ | 23.7$_{(0.932)}$ | 23.6$_{(0.945)}$ |
| MSD | 9.121$_{(0.010)}$ | 8.804$_{(0.008)}$ | 8.743$_{(0.008)}$ | 8.747$_{(0.008)}$ | **8.722**$_{(0.008)}$ |
| Naval | 0.002$_{(0.000)}$ | 0.001$_{(0.000)}$ | 0.001$_{(0.000)}$ | **0.000**$_{(0.000)}$ | **0.000**$_{(0.000)}$ |
| News | **11162**$_{(1153)}$ | **11047**$_{(1106)}$ | **11036**$_{(1118)}$ | **11036**$_{(1116)}$ | **11032**$_{(1118)}$ |
| Obesity | 5.315$_{(0.022)}$ | 3.658$_{(0.033)}$ | **3.572**$_{(0.038)}$ | **3.576**$_{(0.037)}$ | **3.567**$_{(0.037)}$ |
| Power | 3.836$_{(0.045)}$ | 3.017$_{(0.056)}$ | **2.924**$_{(0.065)}$ | **2.941**$_{(0.059)}$ | **2.912**$_{(0.063)}$ |
| Protein | 4.525$_{(0.040)}$ | **3.455**$_{(0.021)}$ | 3.520$_{(0.019)}$ | 3.512$_{(0.017)}$ | 3.493$_{(0.018)}$ |
| STAR | 233$_{(2.388)}$ | **229**$_{(2.076)}$ | **229**$_{(1.850)}$ | **228**$_{(1.985)}$ | **228**$_{(1.857)}$ |
| Superconductor | **0.170**$_{(0.101)}$ | 0.425$_{(0.091)}$ | 0.463$_{(0.087)}$ | 0.427$_{(0.088)}$ | 0.419$_{(0.089)}$ |
| Synthetic | **10.2**$_{(0.068)}$ | **10.1**$_{(0.072)}$ | **10.2**$_{(0.072)}$ | **10.1**$_{(0.073)}$ | **10.1**$_{(0.073)}$ |
| Wave | 13537$_{(32.7)}$ | 7895$_{(86.0)}$ | 4803$_{(37.5)}$ | 4899$_{(55.0)}$ | **4020**$_{(33.5)}$ |
| Wine | 0.693$_{(0.010)}$ | **0.603**$_{(0.010)}$ | 0.626$_{(0.010)}$ | **0.596**$_{(0.012)}$ | **0.598**$_{(0.011)}$ |
| Yacht | 0.761$_{(0.106)}$ | 0.809$_{(0.103)}$ | **0.677**$_{(0.124)}$ | **0.668**$_{(0.125)}$ | **0.645**$_{(0.124)}$ |
| IBUG W-T-L | 16-5-1 | 13-8-1 | 6-16-0 | - | 2-13-7 |
| IBUG+CBU W-T-L | 18-3-1 | 12-9-1 | 16-6-0 | 7-13-2 | - |

Table 8: Probabilistic (NLL ↓) performance for each method on each dataset.

| Dataset | NGBoost | PGBM | CBU | IBUG | IBUG+CBU |
|---|---|---|---|---|---|
| Ames | 11.3$_{(0.018)}$ | 11.3$_{(0.029)}$ | 11.9$_{(0.140)}$ | **11.2**$_{(0.030)}$ | 11.5$_{(0.092)}$ |
| Bike | 1.942$_{(0.024)}$ | 1.929$_{(0.078)}$ | **1.184**$_{(0.034)}$ | 1.886$_{(0.056)}$ | 1.382$_{(0.042)}$ |
| California | 0.545$_{(0.007)}$ | 0.580$_{(0.005)}$ | 0.524$_{(0.004)}$ | 0.477$_{(0.010)}$ | **0.437**$_{(0.016)}$ |
| Communities | **-0.697**$_{(0.045)}$ | **-0.666**$_{(0.034)}$ | **-0.614**$_{(0.109)}$ | **-0.639**$_{(0.135)}$ | **-0.665**$_{(0.116)}$ |
| Concrete | 3.043$_{(0.030)}$ | 2.802$_{(0.083)}$ | **2.766**$_{(0.086)}$ | 2.980$_{(0.146)}$ | **2.695**$_{(0.060)}$ |
| Energy | 0.604$_{(0.192)}$ | **0.322**$_{(0.182)}$ | **0.406**$_{(0.116)}$ | 1.644$_{(0.514)}$ | 0.658$_{(0.165)}$ |
| Facebook | **2.102**$_{(0.026)}$ | 3.116$_{(0.077)}$ | 2.574$_{(0.191)}$ | 2.175$_{(0.067)}$ | 2.276$_{(0.140)}$ |
| Kin8nm | -0.414$_{(0.007)}$ | -0.774$_{(0.034)}$ | -0.772$_{(0.008)}$ | **-0.841**$_{(0.008)}$ | **-0.847**$_{(0.010)}$ |
| Life | 2.163$_{(0.029)}$ | 1.943$_{(0.033)}$ | 1.932$_{(0.079)}$ | 1.858$_{(0.033)}$ | **1.783**$_{(0.041)}$ |
| MEPS | **3.722**$_{(0.050)}$ | 3.902$_{(0.049)}$ | **3.699**$_{(0.038)}$ | 3.793$_{(0.052)}$ | **3.675**$_{(0.041)}$ |
| MSD | 3.454$_{(0.002)}$ | 3.571$_{(0.002)}$ | 3.415$_{(0.001)}$ | 3.415$_{(0.002)}$ | **3.393**$_{(0.001)}$ |
| Naval | -5.408$_{(0.007)}$ | -5.064$_{(0.338)}$ | -6.141$_{(0.013)}$ | -6.208$_{(0.010)}$ | **-6.284**$_{(0.007)}$ |
| News | 10.9$_{(0.268)}$ | **10.7**$_{(0.339)}$ | **10.6**$_{(0.205)}$ | **10.6**$_{(0.208)}$ | **10.6**$_{(0.192)}$ |
| Obesity | 2.940$_{(0.003)}$ | 2.604$_{(0.015)}$ | **2.439**$_{(0.009)}$ | 2.646$_{(0.009)}$ | 2.515$_{(0.010)}$ |
| Power | 2.752$_{(0.032)}$ | **2.518**$_{(0.021)}$ | 2.538$_{(0.019)}$ | 2.575$_{(0.036)}$ | **2.514**$_{(0.017)}$ |
| Protein | 2.840$_{(0.014)}$ | 2.661$_{(0.005)}$ | 2.553$_{(0.009)}$ | 2.653$_{(0.054)}$ | **2.516**$_{(0.010)}$ |
| STAR | 6.869$_{(0.013)}$ | 6.866$_{(0.012)}$ | **6.866**$_{(0.014)}$ | **6.853**$_{(0.008)}$ | **6.852**$_{(0.009)}$ |
| Superconductor | **12.2**$_{(13.1)}$ | **0.035**$_{(0.095)}$ | **-0.014**$_{(0.078)}$ | 0.783$_{(0.181)}$ | 0.108$_{(0.036)}$ |
| Synthetic | 3.745$_{(0.007)}$ | **3.742**$_{(0.006)}$ | **3.741**$_{(0.008)}$ | **3.738**$_{(0.007)}$ | **3.738**$_{(0.007)}$ |
| Wave | 10.7$_{(0.002)}$ | 10.3$_{(0.021)}$ | **9.675**$_{(0.003)}$ | 10.5$_{(0.030)}$ | 9.760$_{(0.046)}$ |
| Wine | 1.025$_{(0.013)}$ | 0.952$_{(0.020)}$ | 1.025$_{(0.028)}$ | **0.910**$_{(0.016)}$ | 0.933$_{(0.012)}$ |
| Yacht | 0.905$_{(0.232)}$ | **0.357**$_{(0.162)}$ | 0.951$_{(0.252)}$ | 1.799$_{(1.307)}$ | 0.840$_{(0.310)}$ |
| IBUG W-T-L | 12-10-0 | 7-11-4 | 5-11-6 | - | 2-8-12 |
| IBUG+CBU W-T-L | 15-6-1 | 10-10-2 | 13-6-3 | 12-8-2 | - |

---

[3]https://uncertainty-toolbox.github.io/

**Check and interval scores.** Tables 9 and 10 show results when measuring performance with two additional proper scoring rules [29], *check score* (a.k.a. "pinball loss") and *interval score* (evaluation using a pair of quantiles with expected coverage). Under these additional metrics, IBUG+CBU still outperform all other approaches.

Table 9: Probabilistic (check score a.k.a. "pinball loss" $\downarrow$) performance.

| Dataset | NGBoost | PGBM | CBU | IBUG | IBUG+CBU |
|---|---|---|---|---|---|
| Ames | $19358_{(276)}$ | $5487_{(179)}$ | $5551_{(167)}$ | $\mathbf{5266}_{(185)}$ | $\mathbf{5145}_{(186)}$ |
| Bike | $6.264_{(0.482)}$ | $0.597_{(0.020)}$ | $0.420_{(0.018)}$ | $0.490_{(0.024)}$ | $\mathbf{0.386}_{(0.016)}$ |
| California | $8e+10_{(8e+10)}$ | $0.112_{(4e-04)}$ | $0.110_{(4e-04)}$ | $0.107_{(5e-04)}$ | $\mathbf{0.104}_{(4e-04)}$ |
| Communities | $0.034_{(0.001)}$ | $0.034_{(1e-03)}$ | $0.034_{(9e-04)}$ | $\mathbf{0.033}_{(9e-04)}$ | $\mathbf{0.033}_{(9e-04)}$ |
| Concrete | $1.722_{(0.092)}$ | $0.972_{(0.043)}$ | $\mathbf{0.902}_{(0.039)}$ | $0.932_{(0.049)}$ | $\mathbf{0.878}_{(0.041)}$ |
| Energy | $0.262_{(0.022)}$ | $\mathbf{0.074}_{(0.003)}$ | $0.099_{(0.005)}$ | $\mathbf{0.072}_{(0.005)}$ | $0.079_{(0.004)}$ |
| Facebook | $2.024_{(0.049)}$ | $1.788_{(0.047)}$ | $1.617_{(0.030)}$ | $1.551_{(0.033)}$ | $\mathbf{1.502}_{(0.035)}$ |
| Kin8nm | $0.048_{(3e-04)}$ | $0.031_{(5e-04)}$ | $0.029_{(3e-04)}$ | $\mathbf{0.026}_{(3e-04)}$ | $\mathbf{0.026}_{(3e-04)}$ |
| Life | $1.462_{(0.739)}$ | $0.411_{(0.014)}$ | $0.389_{(0.012)}$ | $0.400_{(0.011)}$ | $\mathbf{0.368}_{(0.011)}$ |
| MEPS | $\mathbf{2.779}_{(0.098)}$ | $3.246_{(0.046)}$ | $3.050_{(0.055)}$ | $3.100_{(0.057)}$ | $3.033_{(0.056)}$ |
| MSD | $2.283_{(0.003)}$ | $2.310_{(0.002)}$ | $2.203_{(0.002)}$ | $2.226_{(0.002)}$ | $\mathbf{2.195}_{(0.002)}$ |
| Naval | $0.002_{(3e-05)}$ | $2e-04_{(2e-05)}$ | $2e-04_{(2e-06)}$ | $1e-04_{(1e-06)}$ | $\mathbf{1e-04}_{(8e-07)}$ |
| News | $\mathbf{1102}_{(23.7)}$ | $1188_{(26.3)}$ | $1181_{(26.2)}$ | $1280_{(20.5)}$ | $1198_{(26.0)}$ |
| Obesity | $1.620_{(0.014)}$ | $0.939_{(0.011)}$ | $\mathbf{0.879}_{(0.009)}$ | $0.941_{(0.010)}$ | $0.894_{(0.009)}$ |
| Power | $1.063_{(0.012)}$ | $0.773_{(0.010)}$ | $\mathbf{0.744}_{(0.011)}$ | $0.778_{(0.011)}$ | $\mathbf{0.743}_{(0.011)}$ |
| Protein | $2739_{(2730)}$ | $0.920_{(0.006)}$ | $0.902_{(0.005)}$ | $0.900_{(0.004)}$ | $\mathbf{0.880}_{(0.004)}$ |
| STAR | $66.6_{(0.803)}$ | $\mathbf{65.9}_{(0.697)}$ | $65.7_{(0.647)}$ | $65.4_{(0.613)}$ | $65.4_{(0.605)}$ |
| Superconductor | $1.215_{(0.014)}$ | $\mathbf{0.064}_{(0.002)}$ | $0.076_{(0.002)}$ | $0.077_{(0.003)}$ | $\mathbf{0.064}_{(0.002)}$ |
| Synthetic | $2.918_{(0.021)}$ | $\mathbf{2.897}_{(0.020)}$ | $\mathbf{2.898}_{(0.020)}$ | $\mathbf{2.894}_{(0.020)}$ | $\mathbf{2.894}_{(0.020)}$ |
| Wave | $2.9e+05_{(446)}$ | $1964_{(37.3)}$ | $1186_{(5.194)}$ | $1350_{(8.028)}$ | $\mathbf{1023}_{(4.813)}$ |
| Wine | $0.194_{(0.002)}$ | $\mathbf{0.163}_{(0.003)}$ | $0.170_{(0.003)}$ | $\mathbf{0.162}_{(0.003)}$ | $\mathbf{0.162}_{(0.003)}$ |
| Yacht | $0.594_{(0.080)}$ | $\mathbf{0.147}_{(0.021)}$ | $\mathbf{0.142}_{(0.024)}$ | $\mathbf{0.139}_{(0.024)}$ | $\mathbf{0.128}_{(0.023)}$ |
| IBUG W-T-L | 17-3-2 | 11-9-2 | 9-5-8 | - | 1-6-15 |
| IBUG+CBU W-T-L | 17-3-2 | 15-6-1 | 18-2-2 | 15-6-1 | - |

Table 10: Probabilistic (interval score $\downarrow$) performance.

| Dataset | NGBoost | PGBM | CBU | IBUG | IBUG+CBU |
|---|---|---|---|---|---|
| Ames | $2.0e+05_{(3492)}$ | $59165_{(1952)}$ | $66337_{(2499)}$ | $\mathbf{57219}_{(1941)}$ | $\mathbf{55551}_{(1994)}$ |
| Bike | $66.4_{(6.425)}$ | $7.048_{(0.411)}$ | $\mathbf{4.270}_{(0.136)}$ | $6.775_{(0.324)}$ | $\mathbf{4.263}_{(0.191)}$ |
| California | $1e+12_{(1e+12)}$ | $1.257_{(0.008)}$ | $1.168_{(0.008)}$ | $1.230_{(0.020)}$ | $\mathbf{1.119}_{(0.006)}$ |
| Communities | $0.361_{(0.013)}$ | $0.366_{(0.012)}$ | $0.352_{(0.011)}$ | $\mathbf{0.343}_{(0.013)}$ | $\mathbf{0.339}_{(0.011)}$ |
| Concrete | $17.2_{(0.917)}$ | $11.1_{(0.698)}$ | $10.3_{(0.491)}$ | $12.1_{(0.800)}$ | $\mathbf{10.1}_{(0.523)}$ |
| Energy | $2.711_{(0.198)}$ | $\mathbf{0.814}_{(0.050)}$ | $0.998_{(0.067)}$ | $0.912_{(0.083)}$ | $\mathbf{0.819}_{(0.064)}$ |
| Facebook | $28.4_{(0.909)}$ | $26.8_{(1.125)}$ | $21.6_{(0.692)}$ | $\mathbf{17.4}_{(0.476)}$ | $\mathbf{17.1}_{(0.509)}$ |
| Kin8nm | $0.458_{(0.003)}$ | $0.311_{(0.007)}$ | $0.292_{(0.005)}$ | $0.302_{(0.009)}$ | $\mathbf{0.262}_{(0.005)}$ |
| Life | $17.5_{(9.900)}$ | $5.051_{(0.239)}$ | $4.617_{(0.198)}$ | $5.093_{(0.264)}$ | $4.332_{(0.207)}$ |
| MEPS | $42.2_{(1.973)}$ | $44.3_{(1.294)}$ | $\mathbf{37.7}_{(1.223)}$ | $38.3_{(1.375)}$ | $37.2_{(1.254)}$ |
| MSD | $24.5_{(0.039)}$ | $24.8_{(0.035)}$ | $22.3_{(0.020)}$ | $22.4_{(0.029)}$ | $\mathbf{22.0}_{(0.025)}$ |
| Naval | $0.014_{(3e-04)}$ | $0.003_{(3e-04)}$ | $0.002_{(2e-05)}$ | $0.001_{(3e-05)}$ | $\mathbf{0.001}_{(1e-05)}$ |
| News | $\mathbf{16557}_{(519)}$ | $16242_{(556)}$ | $16166_{(580)}$ | $18694_{(373)}$ | $16426_{(551)}$ |
| Obesity | $15.5_{(0.153)}$ | $9.731_{(0.125)}$ | $\mathbf{8.747}_{(0.083)}$ | $10.7_{(0.139)}$ | $9.162_{(0.086)}$ |
| Power | $10.6_{(0.136)}$ | $8.146_{(0.122)}$ | $\mathbf{7.837}_{(0.165)}$ | $8.512_{(0.152)}$ | $\mathbf{7.803}_{(0.156)}$ |
| Protein | $36689_{(36570)}$ | $10.1_{(0.149)}$ | $9.277_{(0.062)}$ | $9.322_{(0.045)}$ | $\mathbf{8.853}_{(0.052)}$ |
| STAR | $642_{(7.014)}$ | $637_{(6.564)}$ | $\mathbf{636}_{(6.131)}$ | $630_{(4.545)}$ | $630_{(4.968)}$ |
| Superconductor | $12.0_{(0.133)}$ | $0.776_{(0.023)}$ | $0.755_{(0.030)}$ | $1.150_{(0.060)}$ | $\mathbf{0.692}_{(0.033)}$ |
| Synthetic | $28.4_{(0.228)}$ | $28.1_{(0.188)}$ | $28.1_{(0.211)}$ | $28.0_{(0.197)}$ | $28.0_{(0.199)}$ |
| Wave | $3e+06_{(3727)}$ | $20256_{(323)}$ | $11748_{(55.8)}$ | $16669_{(117)}$ | $\mathbf{10569}_{(47.4)}$ |
| Wine | $1.930_{(0.023)}$ | $\mathbf{1.723}_{(0.032)}$ | $1.793_{(0.035)}$ | $\mathbf{1.716}_{(0.030)}$ | $\mathbf{1.692}_{(0.031)}$ |
| Yacht | $5.798_{(0.808)}$ | $\mathbf{1.621}_{(0.248)}$ | $1.796_{(0.419)}$ | $1.955_{(0.433)}$ | $\mathbf{1.619}_{(0.406)}$ |
| IBUG W-T-L | 18-3-1 | 8-9-5 | 4-8-10 | - | 0-6-16 |
| IBUG+CBU W-T-L | 18-4-0 | 16-6-0 | 16-4-2 | 16-6-0 | - |

**Calibration error.** Table 11 shows the average MACE (mean absolute calibration error) and sharpness scores. Sharpness quantifies the average of the standard deviations and thus does not depend on the actual ground-truth label; therefore, MACE and sharpness are shown together, with better methods having both low calibration error and low sharpness scores.

We observe that NGBoost is particularly well-calibrated, but lacks sharpness, meaning the prediction intervals of NGBoost are generally too wide. PGBM tends to have very sharp prediction intervals, but high calibration error. In contrast, CBU tends to achieve both low calibration error and high sharpness in relation to the other methods. However, these results are with variance calibration (§3.2), which we note has a significant impact on the CBU approach. For example, the median improvement in MACE score (over datasets) for CBU when using variance calibration vs. without is greater than 3x.

Table 11: Probabilistic (MACE ↓ / sharpness ↓) performance. Standard errors are omitted for brevity.

| Dataset | NGBoost | PGBM | CBU | IBUG | IBUG+CBU |
|---|---|---|---|---|---|
| Ames | 0.082/74148 | **0.040/18432** | 0.073/18867 | 0.068/23186 | 0.063/19791 |
| Bike | 0.070/190 | 0.140/2.077 | **0.045**/2.136 | 0.096/**1.272** | 0.051/1.595 |
| California | **0.014**/3e+13 | 0.053/**0.344** | 0.021/0.367 | 0.089/0.382 | 0.037/0.364 |
| Communities | 0.039/0.129 | 0.067/**0.120** | 0.051/0.136 | **0.035**/0.133 | 0.048/0.133 |
| Concrete | 0.056/6.889 | 0.068/3.002 | 0.096/3.177 | 0.115/**2.503** | **0.054**/2.708 |
| Energy | 0.127/1.497 | 0.093/0.252 | 0.054/0.373 | 0.103/**0.249** | **0.053**/0.296 |
| Facebook | 0.094/9.171 | 0.206/**4.309** | 0.072/7.332 | **0.061**/18.9 | 0.091/12.5 |
| Kin8nm | 0.020/0.182 | 0.037/0.108 | **0.020**/0.096 | 0.126/**0.071** | 0.045/0.081 |
| Life | **0.039**/111 | 0.069/**1.103** | 0.079/1.189 | 0.115/1.401 | 0.069/1.216 |
| MEPS | **0.030/6.680** | 0.074/8.200 | 0.119/14.1 | 0.086/17.2 | 0.106/15.3 |
| MSD | **0.007**/7.749 | 0.036/**7.436** | 0.012/8.137 | 0.039/9.088 | 0.031/8.519 |
| Naval | **0.032**/0.006 | 0.279/1e-03 | 0.048/6e-04 | 0.059/**5e-04** | 0.086/5e-04 |
| News | 0.104/**2170** | **0.085**/3289 | 0.101/2975 | 0.202/4803 | 0.109/3498 |
| Obesity | 0.012/5.996 | 0.065/3.451 | **0.006**/3.102 | 0.095/2.957 | 0.043/**2.956** |
| Power | 0.020/3.761 | 0.026/2.558 | **0.018/2.299** | 0.030/3.328 | 0.019/2.729 |
| Protein | 0.029/2e+06 | 0.076/**2.823** | 0.037/3.144 | **0.016**/3.977 | 0.046/3.498 |
| STAR | 0.025/248 | 0.031/250 | 0.030/**242** | **0.023**/245 | 0.025/243 |
| Superconductor | 0.074/7.993 | 0.102/0.240 | **0.028**/0.322 | 0.205/**0.208** | 0.041/0.240 |
| Synthetic | **0.012**/10.4 | 0.023/10.9 | 0.019/**10.4** | 0.012/10.4 | 0.014/10.4 |
| Wave | 0.129/1e+06 | 0.018/6403 | **0.007/4310** | 0.089/6496 | 0.042/5127 |
| Wine | **0.017**/0.694 | 0.070/**0.540** | 0.027/0.575 | 0.091/0.643 | 0.061/0.600 |
| Yacht | 0.115/4.057 | 0.174/0.690 | 0.098/0.508 | 0.124/**0.371** | **0.078**/0.412 |

## B.4 Runtime

Tables 12 and 13 provide detailed runtime results for each method. Results are averaged over 10 folds, and standard deviations are shown in subscripted parentheses; lower is better. The last row in each table shows the Geometric mean over all datasets.

Table 12: Total train (including tuning) time (in seconds).

| Dataset | NGBoost | PGBM | CBU | IBUG |
|---|---|---|---|---|
| Ames | $\mathbf{417}_{(587)}$ | $1.4e{+}05_{(25170)}$ | $23181_{(258)}$ | $22264_{(796)}$ |
| Bike | $\mathbf{195}_{(143)}$ | $58246_{(8207)}$ | $23207_{(239)}$ | $22417_{(794)}$ |
| California | $\mathbf{315}_{(90.8)}$ | $16958_{(1173)}$ | $23141_{(253)}$ | $22530_{(649)}$ |
| Communities | $\mathbf{38.0}_{(21.2)}$ | $24491_{(4246)}$ | $23023_{(260)}$ | $22429_{(483)}$ |
| Concrete | $\mathbf{57.1}_{(22.9)}$ | $4130_{(3621)}$ | $22953_{(265)}$ | $22402_{(577)}$ |
| Energy | $\mathbf{35.3}_{(33.0)}$ | $2706_{(601)}$ | $22783_{(278)}$ | $22423_{(602)}$ |
| Facebook | $\mathbf{731}_{(659)}$ | $3.5e{+}05_{(58586)}$ | $23061_{(310)}$ | $23145_{(517)}$ |
| Kin8nm | $\mathbf{77.8}_{(39.6)}$ | $10489_{(2181)}$ | $23142_{(296)}$ | $22694_{(529)}$ |
| Life | $\mathbf{105}_{(87.4)}$ | $83814_{(25313)}$ | $23082_{(273)}$ | $20531_{(7050)}$ |
| MEPS | $\mathbf{351}_{(477)}$ | $2.3e{+}05_{(41039)}$ | $23139_{(327)}$ | $20491_{(7004)}$ |
| MSD | $\mathbf{11720}_{(1022)}$ | $2.2e{+}05_{(34478)}$ | $23972_{(258)}$ | $52760_{(16670)}$ |
| Naval | $\mathbf{847}_{(1804)}$ | $38882_{(11481)}$ | $23133_{(210)}$ | $20607_{(7059)}$ |
| News | $\mathbf{2275}_{(138)}$ | $2.4e{+}05_{(60642)}$ | $22960_{(258)}$ | $22492_{(448)}$ |
| Obesity | $\mathbf{1569}_{(2208)}$ | $3.2e{+}05_{(64847)}$ | $23169_{(259)}$ | $21040_{(7086)}$ |
| Power | $\mathbf{107}_{(53.0)}$ | $12556_{(1459)}$ | $23042_{(338)}$ | $22445_{(298)}$ |
| Protein | $\mathbf{1430}_{(2298)}$ | $40132_{(5779)}$ | $23043_{(277)}$ | $22865_{(344)}$ |
| STAR | $\mathbf{17.0}_{(5.788)}$ | $20797_{(3481)}$ | $22852_{(263)}$ | $22074_{(432)}$ |
| Superconductor | $\mathbf{215}_{(29.3)}$ | $2.1e{+}05_{(42859)}$ | $23291_{(706)}$ | $22503_{(426)}$ |
| Synthetic | $\mathbf{439}_{(351)}$ | $1.0e{+}05_{(15729)}$ | $23068_{(333)}$ | $22394_{(532)}$ |
| Wave | $\mathbf{3487}_{(342)}$ | $1.0e{+}05_{(16204)}$ | $23394_{(173)}$ | $44282_{(16994)}$ |
| Wine | $\mathbf{33.1}_{(19.2)}$ | $15067_{(2804)}$ | $20942_{(7191)}$ | $22269_{(451)}$ |
| Yacht | $\mathbf{51.3}_{(41.8)}$ | $1965_{(87.7)}$ | $22915_{(239)}$ | $22184_{(433)}$ |
| Geo. mean | $\mathbf{265}$ | $43604$ | $23017$ | $23726$ |

Table 13: Average prediction time per text example (in milliseconds).

| Dataset | NGBoost | PGBM | CBU | IBUG |
|---|---|---|---|---|
| Ames | $5.583_{(5.778)}$ | $9.505_{(2.426)}$ | $\mathbf{0.066}_{(0.010)}$ | $4.851_{(2.766)}$ |
| Bike | $\mathbf{0.514}_{(0.815)}$ | $7.705_{(8.198)}$ | $\mathbf{0.010}_{(0.002)}$ | $61.6_{(21.1)}$ |
| California | $0.243_{(0.082)}$ | $\mathbf{5.659}_{(9.562)}$ | $\mathbf{0.004}_{(0.001)}$ | $23.4_{(5.265)}$ |
| Communities | $0.393_{(0.170)}$ | $11.5_{(0.941)}$ | $\mathbf{0.027}_{(0.003)}$ | $1.803_{(1.118)}$ |
| Concrete | $2.154_{(0.884)}$ | $\mathbf{44.8}_{(57.5)}$ | $\mathbf{0.043}_{(0.019)}$ | $1.876_{(0.726)}$ |
| Energy | $1.830_{(1.369)}$ | $32.9_{(12.1)}$ | $\mathbf{0.053}_{(0.027)}$ | $1.135_{(0.300)}$ |
| Facebook | $0.533_{(0.469)}$ | $\mathbf{5.148}_{(7.166)}$ | $\mathbf{0.024}_{(0.004)}$ | $105_{(62.4)}$ |
| Kin8nm | $0.194_{(0.107)}$ | $168_{(89.7)}$ | $\mathbf{0.008}_{(0.002)}$ | $4.713_{(0.454)}$ |
| Life | $1.466_{(1.171)}$ | $31.5_{(28.1)}$ | $\mathbf{0.064}_{(0.037)}$ | $6.376_{(1.275)}$ |
| MEPS | $0.465_{(0.121)}$ | $\mathbf{9.510}_{(23.6)}$ | $\mathbf{0.005}_{(0.002)}$ | $8.845_{(7.866)}$ |
| MSD | $1.712_{(0.347)}$ | $25.3_{(1.933)}$ | $\mathbf{0.003}_{(7e-04)}$ | $603_{(97.4)}$ |
| Naval | $0.280_{(0.129)}$ | $187_{(253)}$ | $\mathbf{0.010}_{(0.007)}$ | $41.9_{(22.3)}$ |
| News | $0.577_{(0.100)}$ | $0.771_{(0.403)}$ | $\mathbf{0.002}_{(1e-03)}$ | $40.0_{(38.0)}$ |
| Obesity | $0.988_{(0.475)}$ | $10.1_{(6.503)}$ | $\mathbf{0.020}_{(0.003)}$ | $110_{(9.535)}$ |
| Power | $0.252_{(0.115)}$ | $6.904_{(4.256)}$ | $\mathbf{0.007}_{(0.002)}$ | $18.3_{(17.5)}$ |
| Protein | $0.154_{(0.080)}$ | $130_{(73.7)}$ | $\mathbf{0.004}_{(7e-04)}$ | $90.8_{(25.7)}$ |
| STAR | $0.353_{(0.121)}$ | $10.0_{(1.191)}$ | $\mathbf{0.038}_{(0.010)}$ | $0.937_{(0.369)}$ |
| Superconductor | $0.096_{(0.055)}$ | $\mathbf{27.1}_{(79.5)}$ | $\mathbf{0.005}_{(0.002)}$ | $52.6_{(25.8)}$ |
| Synthetic | $\mathbf{0.482}_{(0.541)}$ | $2.461_{(0.524)}$ | $\mathbf{0.009}_{(0.005)}$ | $\mathbf{7.595}_{(10.1)}$ |
| Wave | $\mathbf{1.018}_{(1.339)}$ | $\mathbf{9.116}_{(17.0)}$ | $\mathbf{0.003}_{(3e-04)}$ | $719_{(106)}$ |
| Wine | $0.135_{(0.061)}$ | $280_{(281)}$ | $\mathbf{0.009}_{(0.002)}$ | $6.001_{(1.696)}$ |
| Yacht | $4.192_{(2.669)}$ | $71.5_{(12.1)}$ | $\mathbf{0.124}_{(0.081)}$ | $1.237_{(0.334)}$ |
| Geo. mean | $0.585$ | $18.5$ | $\mathbf{0.013}$ | $15.9$ |

# C   Additional Experiments

In this section, we present additional experimental results.

## C.1   Probabilistic Performance Without Variance Calibration

Tables 14 and 15 show the probabilistic performance of each method *without* variance calibration. Even without variance calibration, IBUG+CBU generally outperforms competing methods. Standard errors are shown in subscripted parentheses.

Table 14: Probabilistic (CRPS ↓) performance *without* variance calibration.

| Dataset | NGBoost | PGBM | CBU | IBUG | IBUG+CBU |
|---|---|---|---|---|---|
| Ames | $38279_{(564)}$ | $12173_{(484)}$ | $11948_{(386)}$ | $\mathbf{10442}_{(373)}$ | $10208_{(392)}$ |
| Bike | $13.9_{(1.856)}$ | $1.274_{(0.054)}$ | $\mathbf{0.835}_{(0.035)}$ | $1.899_{(0.224)}$ | $1.219_{(0.105)}$ |
| California | $2e+11_{(2e+11)}$ | $0.227_{(0.004)}$ | $0.221_{(0.001)}$ | $0.213_{(1e-03)}$ | $\mathbf{0.207}_{(9e-04)}$ |
| Communities | $0.068_{(0.002)}$ | $0.077_{(0.004)}$ | $0.070_{(0.002)}$ | $\mathbf{0.065}_{(0.002)}$ | $\mathbf{0.065}_{(0.002)}$ |
| Concrete | $3.395_{(0.181)}$ | $1.932_{(0.088)}$ | $1.994_{(0.095)}$ | $1.938_{(0.079)}$ | $\mathbf{1.780}_{(0.085)}$ |
| Energy | $0.539_{(0.042)}$ | $\mathbf{0.151}_{(0.007)}$ | $0.207_{(0.010)}$ | $0.481_{(0.041)}$ | $0.293_{(0.022)}$ |
| Facebook | $4.022_{(0.099)}$ | $3.860_{(0.149)}$ | $3.214_{(0.058)}$ | $3.072_{(0.066)}$ | $\mathbf{2.971}_{(0.072)}$ |
| Kin8nm | $0.095_{(6e-04)}$ | $0.069_{(0.000)}$ | $0.063_{(8e-04)}$ | $0.052_{(4e-04)}$ | $\mathbf{0.052}_{(6e-04)}$ |
| Life | $2.897_{(1.465)}$ | $0.836_{(0.035)}$ | $0.852_{(0.030)}$ | $0.794_{(0.022)}$ | $\mathbf{0.739}_{(0.024)}$ |
| MEPS | $\mathbf{5.529}_{(0.196)}$ | $6.725_{(0.126)}$ | $6.050_{(0.109)}$ | $6.146_{(0.113)}$ | $6.022_{(0.114)}$ |
| MSD | $4.525_{(0.005)}$ | $5.767_{(0.006)}$ | $4.364_{(0.004)}$ | $4.410_{(0.005)}$ | $\mathbf{4.342}_{(0.004)}$ |
| Naval | $0.003_{(6e-05)}$ | $0.005_{(0.002)}$ | $3e-04_{(3e-06)}$ | $3e-04_{(2e-06)}$ | $\mathbf{3e-04}_{(2e-06)}$ |
| News | $\mathbf{2476}_{(38.9)}$ | $2628_{(94.9)}$ | $2712_{(59.7)}$ | $2669_{(43.9)}$ | $2593_{(49.7)}$ |
| Obesity | $3.208_{(0.028)}$ | $1.860_{(0.022)}$ | $\mathbf{1.754}_{(0.017)}$ | $1.882_{(0.019)}$ | $1.772_{(0.018)}$ |
| Power | $2.104_{(0.024)}$ | $1.585_{(0.057)}$ | $1.572_{(0.024)}$ | $1.542_{(0.020)}$ | $\mathbf{1.488}_{(0.022)}$ |
| Protein | $5427_{(5409)}$ | $1.932_{(0.014)}$ | $1.822_{(0.010)}$ | $1.785_{(0.008)}$ | $\mathbf{1.740}_{(0.009)}$ |
| STAR | $132_{(1.697)}$ | $157_{(6.908)}$ | $132_{(1.540)}$ | $\mathbf{129}_{(1.225)}$ | $130_{(1.327)}$ |
| Superconductor | $3.200_{(0.031)}$ | $\mathbf{0.134}_{(0.005)}$ | $0.151_{(0.004)}$ | $0.303_{(0.025)}$ | $0.201_{(0.013)}$ |
| Synthetic | $5.778_{(0.043)}$ | $6.946_{(0.242)}$ | $\mathbf{5.769}_{(0.049)}$ | $5.731_{(0.040)}$ | $5.735_{(0.042)}$ |
| Wave | $5.7e+05_{(886)}$ | $4152_{(247)}$ | $\mathbf{2350}_{(10.3)}$ | $4905_{(12.3)}$ | $3112_{(8.952)}$ |
| Wine | $0.385_{(0.005)}$ | $0.383_{(0.015)}$ | $0.355_{(0.007)}$ | $\mathbf{0.322}_{(0.006)}$ | $0.321_{(0.006)}$ |
| Yacht | $1.187_{(0.142)}$ | $\mathbf{0.310}_{(0.056)}$ | $0.291_{(0.050)}$ | $0.644_{(0.068)}$ | $0.394_{(0.042)}$ |
| IBUG W-T-L | 16-4-2 | 12-4-6 | 10-4-8 | - | 0-5-17 |
| IBUG+CBU W-T-L | 17-3-2 | 15-4-3 | 14-2-6 | 17-5-0 | - |

Table 15: Probabilistic (NLL ↓) performance *without* variance calibration.

| Dataset | NGBoost | PGBM | CBU | IBUG | IBUG+CBU |
|---|---|---|---|---|---|
| Ames | $11.3_{(0.023)}$ | $23.7_{(6.813)}$ | $1676_{(1093)}$ | $\mathbf{11.2}_{(0.031)}$ | $11.3_{(0.070)}$ |
| Bike | $1.942_{(0.024)}$ | $11.1_{(4.073)}$ | $\mathbf{1.264}_{(0.080)}$ | $2.958_{(0.106)}$ | $2.386_{(0.091)}$ |
| California | $0.551_{(0.010)}$ | $7.821_{(7.026)}$ | $2.261_{(0.341)}$ | $0.484_{(0.009)}$ | $\mathbf{0.375}_{(0.009)}$ |
| Communities | $\mathbf{-7e-01}_{(0.056)}$ | $20.7_{(7.235)}$ | $2.438_{(1.168)}$ | $-6e-01_{(0.136)}$ | $-4e-01_{(0.269)}$ |
| Concrete | $3.062_{(0.031)}$ | $3.102_{(0.277)}$ | $684_{(358)}$ | $\mathbf{2.848}_{(0.055)}$ | $2.822_{(0.093)}$ |
| Energy | $\mathbf{0.670}_{(0.250)}$ | $0.481_{(0.341)}$ | $6.129_{(3.597)}$ | $1.461_{(0.113)}$ | $1.048_{(0.162)}$ |
| Facebook | $2.099_{(0.026)}$ | $14.7_{(5.523)}$ | $5.147_{(1.834)}$ | $2.195_{(0.070)}$ | $\mathbf{2.044}_{(0.045)}$ |
| Kin8nm | $-4e-01_{(0.007)}$ | $35.0_{(23.1)}$ | $59.5_{(24.2)}$ | $-8e-01_{(0.009)}$ | $\mathbf{-9e-01}_{(0.024)}$ |
| Life | $2.188_{(0.044)}$ | $23.5_{(20.6)}$ | $71.9_{(38.8)}$ | $\mathbf{1.889}_{(0.038)}$ | $1.885_{(0.100)}$ |
| MEPS | $\mathbf{3.732}_{(0.056)}$ | $11.3_{(3.246)}$ | $3.722_{(0.044)}$ | $3.820_{(0.064)}$ | $3.678_{(0.054)}$ |
| MSD | $3.454_{(0.002)}$ | $65.6_{(0.162)}$ | $3.450_{(0.004)}$ | $3.415_{(0.002)}$ | $\mathbf{3.383}_{(0.002)}$ |
| Naval | $-5e+00_{(0.007)}$ | $-4e+00_{(0.357)}$ | $-5e+00_{(0.057)}$ | $-6e+00_{(0.007)}$ | $\mathbf{-6e+00}_{(0.006)}$ |
| News | $\mathbf{10.9}_{(0.335)}$ | $130_{(49.5)}$ | $10.8_{(0.368)}$ | $11.0_{(0.415)}$ | $10.7_{(0.307)}$ |
| Obesity | $2.940_{(0.003)}$ | $2.603_{(0.015)}$ | $\mathbf{2.488}_{(0.009)}$ | $2.646_{(0.009)}$ | $2.493_{(0.009)}$ |
| Power | $2.769_{(0.042)}$ | $11.0_{(8.270)}$ | $5.304_{(0.672)}$ | $\mathbf{2.575}_{(0.036)}$ | $2.569_{(0.057)}$ |
| Protein | $2.841_{(0.015)}$ | $5.299_{(0.268)}$ | $3.291_{(0.042)}$ | $2.747_{(0.123)}$ | $2.531_{(0.028)}$ |
| STAR | $6.872_{(0.015)}$ | $23.2_{(5.203)}$ | $6.989_{(0.040)}$ | $\mathbf{6.853}_{(0.008)}$ | $6.857_{(0.012)}$ |
| Superconductor | $\mathbf{12.1}_{(13.4)}$ | $10.5_{(4.631)}$ | $-6e-03_{(0.093)}$ | $1.151_{(0.111)}$ | $0.602_{(0.094)}$ |
| Synthetic | $3.746_{(0.008)}$ | $27.3_{(6.288)}$ | $3.782_{(0.032)}$ | $\mathbf{3.738}_{(0.007)}$ | $3.744_{(0.010)}$ |
| Wave | $10.7_{(0.002)}$ | $22.2_{(7.985)}$ | $\mathbf{9.679}_{(0.004)}$ | $10.9_{(0.004)}$ | $10.4_{(0.004)}$ |
| Wine | $1.029_{(0.014)}$ | $109_{(24.9)}$ | $578_{(428)}$ | $\mathbf{0.910}_{(0.016)}$ | $0.968_{(0.030)}$ |
| Yacht | $\mathbf{0.904}_{(0.232)}$ | $7.227_{(4.096)}$ | $4.770_{(2.330)}$ | $1.502_{(0.308)}$ | $1.204_{(0.519)}$ |
| IBUG W-T-L | 11-7-4 | 10-10-2 | 8-9-5 | - | 1-10-11 |
| IBUG+CBU W-T-L | 13-7-2 | 11-10-1 | 8-11-3 | 11-10-1 | - |

## C.2 Comparison to $k$-Nearest Neighbors

In this section, we compare IBUG to $k$-nearest neighbors, in which similarity is defined by Euclidean distance. For the nearest-neighbors approach, we tune two different $k$ values, one for estimating the conditional mean, and one for estimating the variance. We also apply standard scaling to the data before training, and denote this method $k$NN in our results. Table 16 shows that IBUG is consistently better than $k$NN in terms of probabilistic performance. However, we note that point predictions from GBRTs is typically better than $k$NNs, thus we also compare IBUG to a variant of $k$NN that uses CatBoost as a base model to estimate the conditional mean.

**Euclidean Distance vs. Affinity.** To test which similarity measure (Euclidean distance or affinity) is more effective, we use the output from CatBoost to model the conditional mean, we then use $k$NN or IBUG to find their respective $k$-nearest training examples to estimate the variance; we denote these methods $k$NN-CB[4] and IBUG-CB. For $k$NN-CB, we also reduce the dimensionality of the data by only using the most important features identified by the CatBoost model;[5] this helps $k$NN-CB combat the curse of dimensionality when computing similarity. Results of this comparison are in Table 16, in which we observe IBUG-CB is always statistically the same or better than $k$NN-CB. These results suggest affinity is a more effective similarity measure than Euclidean distance for uncertainty estimation in GBRTs.

Table 16: Probabilistic (CRPS) performance comparison of IBUG against two different nearest-neighbor models. $k$NN estimates the conditional mean and variance using two different $k$ values; and $k$NN-CB estimates the variance in the same way as $k$NN, but uses the scalar output from the CatBoost model to estimate the conditional mean. Overall, these results suggest affinity is a better measure of similarity than Euclidean distance for uncertainty estimation in GBRTs.

| Dataset | $k$NN | $k$NN-CB | IBUG-CB |
|---|---|---|---|
| Bike | $\mathbf{0.932}_{(0.029)}$ | $\mathbf{0.978}_{(0.049)}$ | $\mathbf{0.974}_{(0.048)}$ |
| California | $0.579_{(0.001)}$ | $0.219_{(1e-03)}$ | $\mathbf{0.213}_{(9e-04)}$ |
| Communities | $0.072_{(0.002)}$ | $\mathbf{0.065}_{(0.002)}$ | $\mathbf{0.065}_{(0.002)}$ |
| Concrete | $4.645_{(0.140)}$ | $\mathbf{1.872}_{(0.085)}$ | $\mathbf{1.849}_{(0.098)}$ |
| Energy | $0.875_{(0.016)}$ | $\mathbf{0.153}_{(0.010)}$ | $\mathbf{0.143}_{(0.009)}$ |
| Facebook | $5.613_{(0.065)}$ | $3.275_{(0.068)}$ | $\mathbf{3.073}_{(0.066)}$ |
| Kin8nm | $0.067_{(7e-04)}$ | $\mathbf{0.051}_{(5e-04)}$ | $\mathbf{0.051}_{(6e-04)}$ |
| Life | $4.738_{(0.078)}$ | $\mathbf{0.785}_{(0.024)}$ | $\mathbf{0.794}_{(0.023)}$ |
| MEPS | $7.283_{(0.220)}$ | $\mathbf{6.181}_{(0.107)}$ | $\mathbf{6.150}_{(0.114)}$ |
| MSD | $5.312_{(0.006)}$ | $4.446_{(0.004)}$ | $\mathbf{4.410}_{(0.005)}$ |
| Naval | $8e-04_{(2e-05)}$ | $3e-04_{(2e-06)}$ | $\mathbf{2e-04}_{(2e-06)}$ |
| News | $2654_{(52.0)}$ | $\mathbf{2597}_{(52.3)}$ | $\mathbf{2545}_{(41.0)}$ |
| Obesity | $5.526_{(0.013)}$ | $\mathbf{1.900}_{(0.043)}$ | $\mathbf{1.866}_{(0.021)}$ |
| Power | $2.074_{(0.022)}$ | $\mathbf{1.553}_{(0.020)}$ | $\mathbf{1.542}_{(0.020)}$ |
| Protein | $3.241_{(0.010)}$ | $\mathbf{1.787}_{(0.008)}$ | $\mathbf{1.784}_{(0.008)}$ |
| STAR | $140_{(1.553)}$ | $\mathbf{129}_{(1.204)}$ | $\mathbf{130}_{(1.214)}$ |
| Superconductor | $3.445_{(0.041)}$ | $\mathbf{0.156}_{(0.006)}$ | $\mathbf{0.153}_{(0.006)}$ |
| Synthetic | $6.136_{(0.047)}$ | $\mathbf{5.735}_{(0.039)}$ | $\mathbf{5.731}_{(0.040)}$ |
| Wave | $11987_{(36.6)}$ | $2700_{(17.0)}$ | $\mathbf{2679}_{(16.0)}$ |
| Wine | $0.445_{(0.004)}$ | $\mathbf{0.322}_{(0.006)}$ | $\mathbf{0.322}_{(0.006)}$ |
| Yacht | $3.354_{(0.408)}$ | $\mathbf{0.275}_{(0.048)}$ | $\mathbf{0.276}_{(0.048)}$ |
| IBUG-CB W-T-L | 21-1-0 | 10-12-0 | - |

---

[4]Again, we apply standard scaling to the data before training $k$NN-CB.

[5]We tune the number of important features to use for $k$NN-CB using values [5, 10, 20].

## C.3 Comparison to Bayesian Additive Regression Trees

BART (Bayesian Additive Regression Trees) takes a Bayesian approach to uncertainty estimation in trees [13]. Although well-grounded theoretically, BART requires expensive sampling techniques such as MCMC (Markov Chain Monte Carlo) to provide approximate solutions.

In this section, we compare IBUG to BART using a popular open-source implementation.[6] However, due to BART's computational complexity, we tune the number of trees for both IBUG and BART using values [10, 50, 100, 200], set the number of chains for BART to 5, and run our comparison using a subset of the datasets in our empirical evaluation consisting of 11 relatively small datasets.

Tables 17 and 18 show IBUG consistently outperforms BART in terms of both probabilistic and point performance.

Table 17: Probabilistic (CRPS ↓) performance comparison between IBUG and BART.

| Dataset | BART | IBUG |
|---|---|---|
| Bike | $4.521_{(0.119)}$ | $\mathbf{0.974}_{(0.048)}$ |
| California | $0.285_{(0.001)}$ | $\mathbf{0.213}_{(9e-04)}$ |
| Communities | $0.072_{(0.002)}$ | $\mathbf{0.065}_{(0.002)}$ |
| Concrete | $3.067_{(0.073)}$ | $\mathbf{1.849}_{(0.098)}$ |
| Energy | $0.402_{(0.023)}$ | $\mathbf{0.143}_{(0.009)}$ |
| Kin8nm | $0.107_{(8e-04)}$ | $\mathbf{0.051}_{(6e-04)}$ |
| Naval | $1e\text{-}03_{(1e-05)}$ | $\mathbf{2e\text{-}04}_{(2e-06)}$ |
| Power | $2.225_{(0.018)}$ | $\mathbf{1.542}_{(0.020)}$ |
| STAR | $134_{(1.493)}$ | $\mathbf{130}_{(1.214)}$ |
| Wine | $0.394_{(0.005)}$ | $\mathbf{0.322}_{(0.006)}$ |
| Yacht | $0.849_{(0.039)}$ | $\mathbf{0.276}_{(0.048)}$ |
| IBUG W-T-L | 11-0-0 | - |

Table 18: Point (RMSE ↓) performance comparison between IBUG and BART.

| Dataset | BART | IBUG |
|---|---|---|
| Bike | $8.396_{(0.273)}$ | $\mathbf{2.826}_{(0.200)}$ |
| California | $0.547_{(0.003)}$ | $\mathbf{0.432}_{(0.001)}$ |
| Communities | $0.137_{(0.004)}$ | $\mathbf{0.133}_{(0.004)}$ |
| Concrete | $5.507_{(0.161)}$ | $\mathbf{3.629}_{(0.183)}$ |
| Energy | $0.685_{(0.039)}$ | $\mathbf{0.264}_{(0.023)}$ |
| Kin8nm | $0.186_{(0.001)}$ | $\mathbf{0.086}_{(8e-04)}$ |
| Naval | $0.002_{(2e-05)}$ | $\mathbf{5e\text{-}04}_{(5e-06)}$ |
| Power | $4.057_{(0.049)}$ | $\mathbf{2.941}_{(0.059)}$ |
| STAR | $234_{(2.479)}$ | $\mathbf{228}_{(1.985)}$ |
| Wine | $0.708_{(0.008)}$ | $\mathbf{0.596}_{(0.012)}$ |
| Yacht | $1.624_{(0.121)}$ | $\mathbf{0.668}_{(0.125)}$ |
| IBUG W-T-L | 11-0-0 | - |

---

[6] https://github.com/JakeColtman/bartpy

## C.4    Different Tree-Sampling Strategies

Figure 5 shows the probabilistic (NLL) performance of IBUG as the number of trees sampled ($\tau$) increases using three different sampling strategies: *uniformly at random*, *first-to-last*, and *last-to-first*.

We observe that sampling trees *last-to-first* often requires sampling all trees in order to achieve the lowest NLL on the test set. When sampling *uniformly at random*, NLL tends to plateau starting around 10%. In contrast, sampling trees *first-to-last* on the Kin8nm, Naval, and Wine datasets requires 5% of the trees or less to result in the same or better NLL than when sampling all trees; these results provide some evidence that trees early in training contribute most, and suggest that sampling trees first-to-last may be most effective at obtaining the best probabilistic performance while sampling the fewest number of trees.

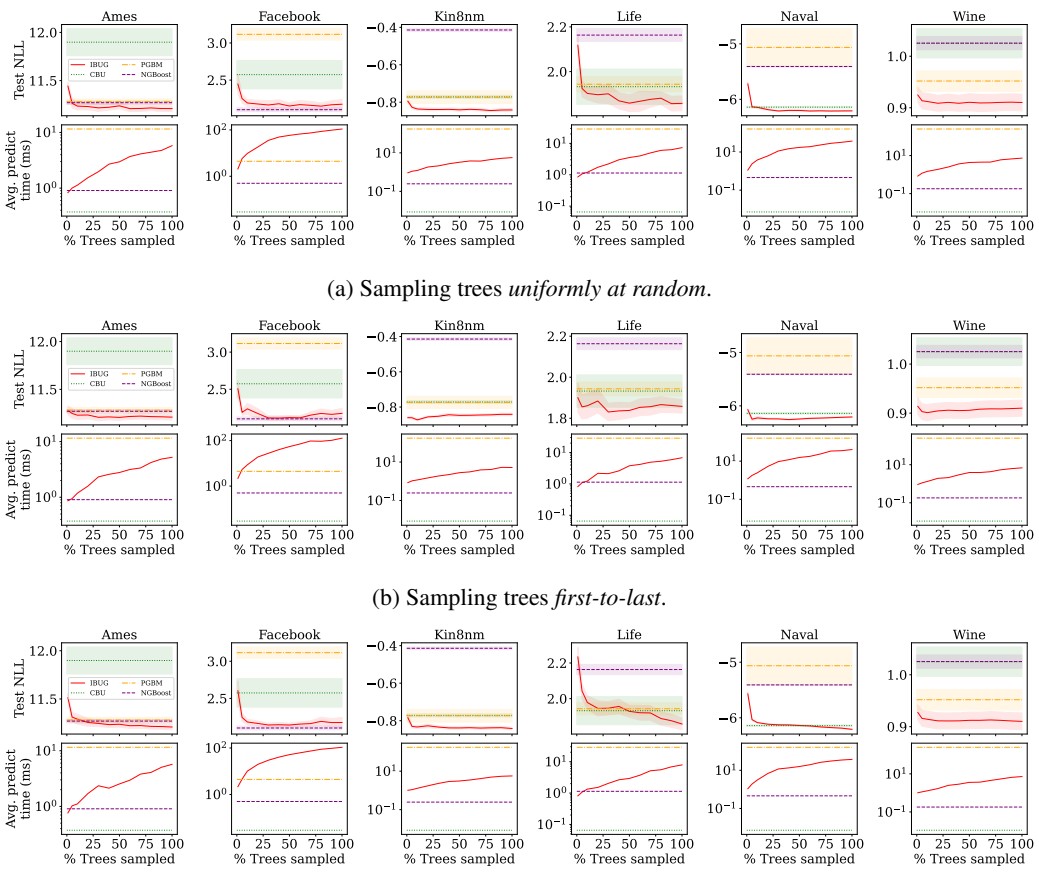

(a) Sampling trees *uniformly at random*.

(b) Sampling trees *first-to-last*.

(c) Sampling trees *last-to-first*.

Figure 5: Probabilistic (NLL) performance (lower is better) and average prediction time (in milliseconds) per test example (lower is better) as a function of $\tau$ for different sampling techniques. *Top*: sample trees uniformly at random, *middle*: sample trees first-to-last (in terms of boosting iteration), *bottom*: sample trees last-to-first. All methods result in similar prediction times; however, *first-to-last* sampling typically provides the best NLL with the fewest number of trees sampled.

## C.5 Leaf Density

Figure 6 shows the average percentage of train instances visited per tree as a function of the total number of training instances for each dataset. We note that for some datasets, CatBoost, LightGBM, and XGBoost induce regression trees with very dense leaves where over half the training instances belong to those leaves. Figure 7 shows average leaf density for each tree in the GBRT.

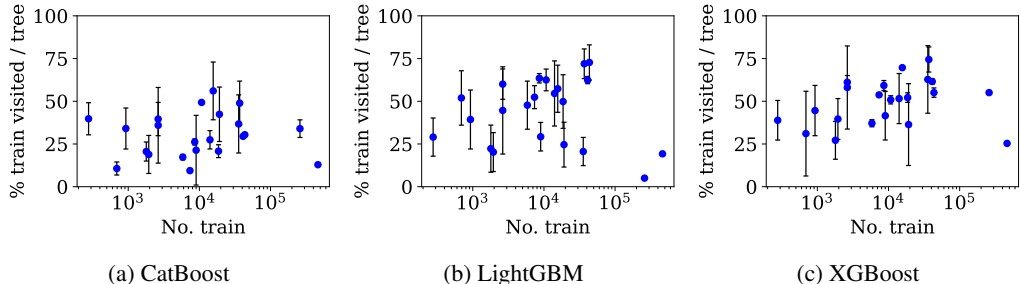

(a) CatBoost  (b) LightGBM  (c) XGBoost

Figure 6: Average percentage of training instances visited per tree while computing affinity vectors on the test set test for each dataset. Results are averaged over all test instances, and error bars represent standard deviation; lower is better. In general, the number of training instances visited per tree is highly dependent on the dataset; and for some datasets, is also highly dependent on the test example (points with large standard deviations).

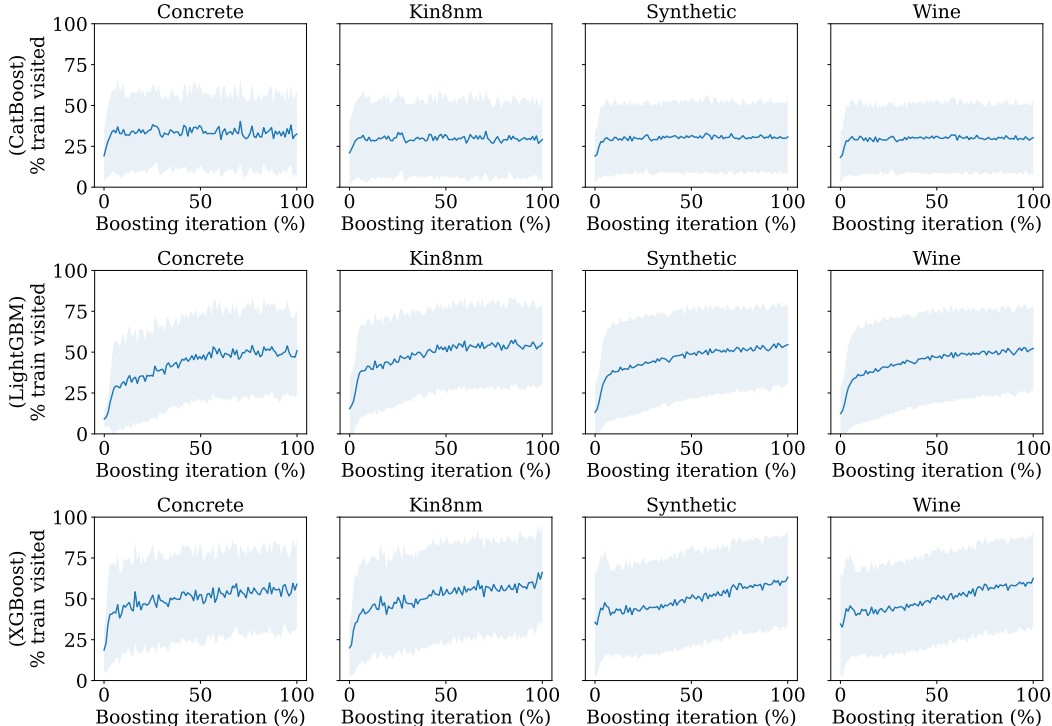

Figure 7: Average percentage of training instances visited at each iteration while computing affinity vectors on the test set for the Concrete, Kin8nm, Synthetic, and Wine datasets. Results are averaged over all test instances, and error bars represent standard deviation; lower is better. For LightGBM and XGBoost, weak learners later in training tend to pool a larger proportion of training instances into fewer leaves; in contrast, CatBoost has less dense leaves and training instances are more equally distributed among the leaves in each tree.