# OpenReview forum: "Instance-Based Uncertainty Estimation for Gradient-Boosted Regression Trees"
_NeurIPS.cc/2022/Conference — NeurIPS 2022 Accept_

### Official Review · Reviewer_TLnP · 2022-06-29

**Rating:** 7
**Confidence:** 3
**Soundness:** 3 good
**Presentation:** 3 good
**Contribution:** 3 good

**Summary:**

The authors develop IBUG, a straightforward approach for producing probabilistic predictions for gradient-boosted regression trees. The approach itself is very simple, but the point the authors are making is that this approach is useful: the authors do a very large amount of computational work showing the potential of their approach. The code the authors provide looks clean and easy-to-use.

The contributions of this paper are largely in engineering a good solution and then implementing that solution in an excellent way: there is no big methodological contribution.

**Questions:**

Would the authors please check their work with respect to Davies & Ghahramani (2014) and document the differences? My impression is that, after considering the work of Davies & Ghahramani (2014), IBUG cannot claim anything methodologically new (except perhaps Section 5.4?), but happy to be corrected.

Would the authors please mention any comparisons they have done between IBUG and the more intricate approaches they mention in Section 6?

Are the authors happy to tone down the claims of "better" with respect to other competing methods?

What is the sensitivity of IBUG to the parameter k? This is the number of nearest neighbors.

**Limitations:**

This is fine.

**Strengths And Weaknesses:**

ORIGINALITY

"Originality" is the weakest aspect of the paper. The authors don't cite Davies & Ghahramani (2014), but the "Affinity Score" in Equation (1) is equivalent to Section 3 in Davies & Ghahramani (2014). Even putting aside Davies & Ghahramani (2014), this paper is not terribly "new" or "novel": nothing in this paper is very surprising.

However, "originality" is often overrated in research: the authors seem to have made a substantial engineering contribution.

Although the authors may have missed Davies & Ghahramani (2014), they otherwise nicely document the literature. So the authors aren't (with the exception of the one paper) over-claiming the originality.

QUALITY

The quality of the engineering is high. The authors have clearly done a lot of work in examining, (i) 22 datasets including 21 standard benchmarks, (ii) 3 performance metrics including NLL, CRPS, and RMSE, (iii) 3 different base models including LightGBM, XGBoost, and CatBoost, (iv) several types of output distributions. I checked the code the authors provided and it looks nice.

CLARITY

The clarity of the paper is good. The paper is well-written and easy to understand. The code is also well-written and easy to understand.

SIGNIFICANCE

The paper has potential to be significant because of the extensive engineering contributions. The (currently anonymous) code is available under an Apache 2.0 license, so my hope is that lots of others will have the opportunity to use the authors' work. As the authors write, there's a fairly big chasm between the ease-of-use of GBRT and the availability of probabilistic models, so closing that gap in an easy-to-use way is nice.

My concerns about significance are two-fold:

(i) The authors only compare to other simple, easy-to-implement metrics for calculating probabilistic prediction on trees. On the one hand, the authors are right in saying that Bayesian models and more complex approaches like BART will not scale well and therefore are unlikely to be used very frequently, but on the other hand it would be nice to have comparisons between the authors' simplified approaches and the more intricate approaches.

(ii) The authors claim "better" performance of IBUG, but I do have some concerns about this because (with trees) the goal should maybe be "different" performance of IBUG to other approaches. Retrofitting GBRT for probabilistic prediction is always going to be a bit of an art since the probabilistic prediction is missing from the beginning and really only added on later.

Although I'm hoping that the authors will address these items in the rebuttal, they're not really show-stopping points. The engineering contribution here is rather extensive.

MINOR

Line 99: euclidean --> Euclidean

---

> ### Author Response · Authors · 2022-08-02
> **Response to Reviewer TLnP**
>
> We thank the reviewer for their thoughtful feedback and appreciate their recognition of the engineering effort put into IBUG.
>
> **Q: Comparison to Davies and Ghahramani (2014)?**
>
> **A:** Yes, the affinity computation is indeed similar to Davies and Ghahramani (2014); our paths then diverge in which we focus on probabilisitic predictions in GBRTs and flexibly modeling the output. We thank the reviewer for bringing this work to our attention, and we will absolutely discuss and cite their work in the main text.
>
> *Davies, Alex, and Zoubin Ghahramani. "The random forest kernel and other kernels for big data from random partitions." arXiv preprint arXiv:1402.4293 (2014).*
>
> **Q: Comparison to BART?**
>
> **A:** We compare IBUG to BART (using the implementation provided at https://github.com/JakeColtman/bartpy) on a subset of the datasets consisting of 13 smaller datasets (we exceeded our computational resources when attempting to run BART on the larger datasets). We tune the number of trees in BART using values [10, 50, 100, 200]; for a fair comparison, we tune the base model for IBUG using the same values. The results are shown below, and we observe that IBUG consistently outperforms BART in terms of point performance and CRPS. IBUG and BART performed similarly in terms of mean absolute calibration error (MACE); however, IBUG had more cases than BART in which the MACE *and* sharpness scores were both low.
>
> ### RMSE
> |Dataset|BART|IBUG|
> | --- | --- | --- |
> |Bike| 8.159| **2.557**|
> |California| 0.536| **0.441**|
> |Communities| **0.134**| 0.134|
> |Concrete| 5.511| **3.917**|
> |Energy| 0.671| **0.322**|
> |Kin8nm| 0.187| **0.108**|
> |Life| 2.678| **1.710**|
> |Naval| 0.002| **0.001**|
> |Power| 4.071| **2.970**|
> |Protein| 4.743| **3.618**|
> |STAR| 234.977| **230.111**|
> |Wine| 0.711| **0.606**|
> |Yacht| 1.447| **0.895**|
> |*IBUG W-L*| 12-1| -|
>
> ### CRPS
> |Dataset|BART|IBUG|
> | --- | --- | --- |
> |Bike| 4.365| **0.959**|
> |California| 0.279| **0.211**|
> |Communities| 0.070| **0.065**|
> |Concrete| 3.088| **2.075**|
> |Energy| 0.384| **0.170**|
> |Kin8nm| 0.107| **0.062**|
> |Life| 1.424| **0.792**|
> |Naval| 0.001| **0.000**|
> |Power| 2.238| **1.533**|
> |Protein| 2.712| **1.808**|
> |STAR| 134.22| **130.24**|
> |Wine| 0.396| **0.324**|
> |Yacht| 0.754| **0.308**|
> |*IBUG W-L*| 13-0| -|
>
> ### MACE/Sharpness
> |Dataset|BART|IBUG|
> | --- | --- | --- |
> |Bike| 0.083/7.334| **0.048**/**2.206**|
> |California| **0.019**/**0.416**| 0.020/0.436|
> |Communities| 0.046/**0.107**| **0.028**/0.129|
> |Concrete| **0.075**/5.530| 0.118/**2.954**|
> |Energy| **0.088**/0.860| 0.163/**0.343**|
> |Kin8nm| 0.081/0.149| **0.055**/**0.143**|
> |Life| 0.054/2.641| **0.038**/**1.599**|
> |Naval| **0.044**/0.002| 0.147/**0.000**|
> |Power| 0.041/3.451| **0.020**/**3.037**|
> |Protein| 0.062/5.925| **0.012**/**3.813**|
> |STAR| 0.045/276.18| **0.022**/**235.51**|
> |Wine| **0.040**/**0.608**| 0.072/0.608|
> |Yacht| **0.118**/1.311| 0.143/**0.818**|
> |*IBUG W-L*| 7-6/10-3| -/-|
> **Performance claims.**
>
> **A:** In general, we describe IBUG as performing similarly or better than existing approaches while offering extra flexibility in terms of model agnosticism and posterior modeling. However, we will make sure our claims convey a notion of *competitive* performance in comparison with existing methods.
>
> **Q: Sensitivity of $k$?**
>
> **A:** We find choosing an appropriate value of $k$ is crucial to achieving good probabilistic performance. However, as our experiments demonstrate, appropriate tuning can lead to effective values of $k$ for a wide range of datasets.
>
> **Typos.**
>
> **A:** We thank the reviewer for spotting this typo.

---

> > ### Comment · Reviewer_TLnP · 2022-08-04
> > **Thanks**
> >
> > Thanks for the response.

---

### Official Review · Reviewer_VfPp · 2022-07-04

**Rating:** 5
**Confidence:** 4
**Soundness:** 3 good
**Presentation:** 4 excellent
**Contribution:** 3 good

**Summary:**

Starting from gradient boosting trees, the paper develops a new method to estimate the conditional distribution P(y|x) for regression problems. The authors propose to estimate P(y|x) in a nearest neighbor approach, using a specific similarity measure. Two instances are considered similar if they end up in the same leaf in many of the trees. Using the neighborhood, the variance or the full conditional distribution P(y|x) can be estimated. The authors also present a calibration procedure for situations where the variance is wrongly estimated during training.

In the experiments, the new method is compared to two specific baselines on 20 tabular datasets, using NLL and CRPS as performance measures.

**Questions:**

Below I am listing potential limitations. I welcome the authors to comment on my remarks.

**Limitations:**

The presented approach is interesting, but I also see some limitations.

The presented approach is a very simple approach with limited novelty. It is in essence a nearest neighbor method with a specific similarity measure. Estimating the conditional distribution by analyzing the neighborhood of a test instance is a well-known approach in nearest neighbor research, so the only novelty here is the similarity measure, which is quite specific. I would not be surprised if this similarity measure outperforms Euclidean distance, because the presented similarity is computed on those features that matter for prediction. Especially for high-dimensional datasets with many irrelevant features this might be an advantage over Euclidean distance. However, in nearest neighbor research, many alternative similarity scores that also provide a solution for the curse of dimensionality have been proposed. I find it a pity that this literature is completely ignored.

To my opinion, the proposed similarity measure has at least one obvious shortcoming: it is a non-continuous function that results in many ties. I am wondering whether this performance measure is able to outperform some simple baselines that also overcome the curse of dimensionality, see e.g. select the most important features based on a variable importance criterion, and compute the Euclidean distance in the resulting lower-dimensional space. Overall, I would have liked to see more theoretical and experimental justification that the proposed similarity measure is the way to go.

The fact that the variance needs to be recalibrated using a validation set lets me conclude that the considered similarity measure leads to a biased estimate of P(y|x). More theoretical insights on what goes wrong would be useful. The experiments show that the new method outperforms some baselines, but that's what 99% of the Neurips submissions claim. These claims are hard to verify, thus some theoretical results would help me to believe that the proposed method is state-of-the-art.

In the experiments, the assumptions for using a paired t-test are not met. Individual numbers are not independent, because the training datasets overlap. If you want to compute p-values, please use the right type of test. See for example Dietterich "Approximate statistical tests for comparing supervised learning algorithms" and follow-up papers on that topic for using the right tests.

NLL and CRPS are hard to interpret as measures for comparing different approaches. To my opinion, checking the validity of a predefined prediction interval is easier to interpret. This measure is commonly used in the conformal prediction for regression literature.

**Strengths And Weaknesses:**

Strengths:
- The paper is well written and easy to follow.
- Uncertainty estimation for gradient-boosted regression trees is an important research problem, because boosted trees usually yield SOTA performance on tabular datasets.
- The experimental evaluation is quite extensive.

Weaknesses:
- The proposed method comes without any theoretical justification. The method is in essence a heuristic.
- I would have liked to see a comparison with nearest neighbor methods that use other similarity measures.

---

> ### Author Response · Authors · 2022-08-02
> **Response to Reviewer VfPp (Part 2)**
>
> **Variance calibration suggests biased uncertainty estimates.**
>
> **A:** We agree that applying variance calibration can help correct for variance estimates that are systematically too large or small. However, variance calibration is not just beneficial for IBUG, but virtually all methods we compare to, with variance calibration helping some methods substantially more than IBUG (e.g., PGBM, also see response to *gTo6*); thus, we view variance calibration as a simple post-processing step that can significantly improve probabilisitic performance for any uncertainty estimator.
>
> **Significance testing.**
>
> **A:** Our protocol of 20 different 90/10 train-test random folds is based on previous work [Duan et al. 2020, Sprangers et al. 2021]. However, we agree with the reviewer that this form of significance testing is somewhat biased and may result in overly optimistic results. The results shown here show wins/losses based on the mean score over the 20 random folds.
>
> **NLL and CRPS are hard to interpret.**
>
> **A:** Negative-log likelihood is a very common metric to use when evaluating probabilisitic predictions in the form of density functions, and CRPS is a popular proper scoring rule that generalizes mean absolute error to probabilistic predictions [Gneiting and Raftery 2007]. However, we also evaluate each method using the *check score* (a.k.a. pinball loss) and interval score (evaluation using a pair of quantiles with expected coverage) using the *Uncertainty Toolbox* [Chung et al. 2021] below. Under these additional metrics, IBUG still performs similarly or better than existing approaches.
>
> ### Check Score (lower is better)
> |Dataset|KNN|KNN-FI|NGBoost|PGBM|IBUG|
> | --- | --- | --- | --- | --- | --- |
> |Ames| 8557| 5444| 19002| 5370| **5274**|
> |Bike| 8.749| 0.486| 6.301| 0.583| **0.418**|
> |California| 0.156| 0.111| 0.129| 0.110| **0.106**|
> |Communities| 0.035| 0.033| 0.033| 0.034| **0.032**|
> |Concrete| 2.268| 0.988| 1.649| **0.956**| 0.982|
> |Energy| 0.965| 0.081| 0.267| 0.082| **0.078**|
> |Facebook| 2.548| 1.658| 2.080| 1.824| **1.543**|
> |Kin8nm| 0.034| **0.030**| 0.048| 0.036| 0.030|
> |Life| 0.455| 0.403| 0.711| 0.413| **0.395**|
> |MEPS| 2.969| 3.207| **2.791**| 3.145| 3.197|
> |MSD| 2.684| 2.303| 2.282| 2.309| **2.206**|
> |Naval| 0.000| 0.000| 0.002| 0.000| **0.000**|
> |News| 1243| 1318| **1081**| 1171| 1223|
> |Obesity| 1.139| 0.032| 2e17| 0.057| **0.029**|
> |Power| 0.988| 0.793| 1.066| 0.774| **0.771**|
> |Protein| 0.869| 0.857| 1.349| 0.935| **0.854**|
> |STAR| 68.924| 66.076| 66.401| 66.149| **65.795**|
> |Superconductor| 0.973| 0.055| 1.227| 0.060| **0.044**|
> |Synthetic| 3.104| **2.912**| 2.948| 2.916| 2.914|
> |Wave| 9754| 2172| 288225| **1974**| 2148|
> |Wine| 0.193| 0.164| 0.195| 0.163| **0.162**|
> |Yacht| 1.758| 0.153| 0.488| **0.121**| 0.152|
> |*IBUG W-L*| 21-1| 20-2| 20-2| 17-5| -|
>
> ### Interval Score (lower is better)
> |Dataset|KNN|KNN-FI|NGBoost|PGBM|IBUG|
> | --- | --- | --- | --- | --- | --- |
> |Ames| 88595| 63262| 194678| 58820| **56298**|
> |Bike| 89.940| 6.840| 65.692| 6.811| **4.960**|
> |California| 1.645| 1.312| 1.357| 1.238| **1.135**|
> |Communities| 0.368| 0.347| 0.351| 0.363| **0.342**|
> |Concrete| 21.932| 11.994| 16.636| **10.648**| 12.152|
> |Energy| 9.419| 1.159| 2.677| **0.957**| 1.048|
> |Facebook| 36.181| 20.641| 29.403| 27.669| **17.447**|
> |Kin8nm| 0.343| 0.327| 0.457| 0.377| **0.320**|
> |Life| 5.213| 4.976| 7.465| 5.138| **4.752**|
> |MEPS| 41.763| **39.778**| 42.406| 41.090| 39.868|
> |MSD| 27.345| 23.977| 24.483| 24.784| **22.338**|
> |Naval| 0.005| **0.001**| 0.014| 0.003| 0.001|
> |News| 18901| 19680| 15985| **15886**| 16906|
> |Obesity| 11.690| 0.398| 2e18| 0.725| **0.351**|
> |Power| 10.214| 8.697| 10.629| **8.105**| 8.251|
> |Protein| 9.619| **9.115**| 13.398| 10.226| 9.136|
> |STAR| 662| 638| 642| 640| **632**|
> |Superconductor| 10.845| 0.861| 12.103| 0.770| **0.515**|
> |Synthetic| 30.103| 28.226| 28.717| 28.259| **28.211**|
> |Wave| 97237| 28368| 2714251| **20332**| 28290|
> |Wine| 1.929| 1.736| 1.938| 1.739| **1.724**|
> |Yacht| 20.211| 1.885| 5.040| **1.312**| 1.664|
> |*IBUG W-L*| 22-0| 18-4| 21-1| 16-6| -|*
>
>
> *Gneiting, Tilmann, and Adrian E. Raftery. "Strictly proper scoring rules, prediction, and estimation." Journal of the American statistical Association 102.477 (2007): 359-378.*
>
> *Chung, Youngseog, et al. "Uncertainty toolbox: an open-source library for assessing, visualizing, and improving uncertainty quantification." arXiv preprint arXiv:2109.10254 (2021). URL: https://uncertainty-toolbox.github.io/.*

---

> ### Author Response · Authors · 2022-08-02
> **Response to Reviewer VfPp (Part 1)**
>
> We are encouraged you find our approach interesting, and we aim to address your major concerns here.
>
> **Q: Comparison to KNN with most important features?**
>
> **A:** The affinity computation in IBUG is an example of a supervised kernel based on the learnt structure of the tree ensemble and we thus expect it to outperform similarity measures based on Euclidean distance. However, we have added an additional comparison to a KNN model that operates on a reduced set of the most important features. First, we apply standard scaling and train a GBRT model to obtain the most important features, then we filter the dataset down to the $\upsilon$ most important features before applying KNN. We treat $\upsilon$ as a hyperparameter and tune $\upsilon$ using values [5, 10, 20]; we denote this method KNN-FI. To estimate the mean and variance, we use the GBRT prediction as the conditional mean, and use KNN-FI to identify the $k$-nearest neighbors (in the reduced feature space) to estimate the variance for each prediction.
>
> Results are shown below, and we observe KNN-FI is able to achieve better point and probabilistic performance than standard KNN. However, IBUG generally outperforms KNN-FI in terms of CRPS and MACE (mean absolute calibration error)/sharpness, demonstrating the effectiveness of a supervised tree-based kernel over a similarity measure like Euclidean distance. We note that KNN achieves good MACE scores but poor sharpness scores, meaning the variance of the KNN predictions are generally too wide. We thank the reviewer for their suggestion and will add this comparison into the paper.
>
> ### RMSE
> |Dataset|KNN|KNN-FI|IBUG|
> | --- | --- | --- | --- |
> |Ames| 35311| **22229**| 22349|
> |Bike| 35.8| 2.903| **2.379**|
> |California| 0.624| 0.436| **0.436**|
> |Communities| 0.142| **0.132**| 0.132|
> |Concrete| 8.433| 3.776| **3.751**|
> |Energy| 3.663| **0.305**| 0.306|
> |Facebook| 30.61| 20.43| **20.38**|
> |Kin8nm| 0.117| **0.102**| 0.103|
> |Life| 2.048| 1.679| **1.678**|
> |MEPS| 24.78| 24.12| **23.85**|
> |MSD| 10.174| **8.778**| 8.780|
> |Naval| 0.002| **0.000**| 0.000|
> |News| **11027**| 11053| 11051|
> |Obesity| 4.276| 0.165| **0.160**|
> |Power| 3.735| 2.965| **2.950**|
> |Protein| 3.811| 3.423| **3.420**|
> |STAR| 240.34| 230.47| **230.31**|
> |Superconductor| 5.321| 0.374| **0.357**|
> |Synthetic| 10.891| **10.197**| 10.209|
> |Wave| 41223| 7615| **7493**|
> |Wine| 0.696| 0.603| **0.601**|
> |Yacht| 8.600| **0.890**| 0.899|
> |*KNN-FI W-L*| 21-1| -| 8-14|
> |*IBUG W-L*| 21-1| 14-8| -|
>
> ### CRPS
> |Dataset|KNN|KNN-FI|IBUG|
> | --- | --- | --- | --- |
> |Ames| 16951| 10793| **10448**|
> |Bike| 17.330| 0.966| **0.829**|
> |California| 0.310| 0.221| **0.210**|
> |Communities| 0.070| 0.065| **0.064**|
> |Concrete| 4.491| 1.960| **1.948**|
> |Energy| 1.911| 0.161| **0.155**|
> |Facebook| 5.064| 3.289| **3.059**|
> |Kin8nm| 0.067| **0.060**| 0.060|
> |Life| 0.902| 0.799| **0.782**|
> |MEPS| **5.900**| 6.362| 6.342|
> |MSD| 5.317| 4.562| **4.369**|
> |Naval| 0.001| 0.000| **0.000**|
> |News| 2473| 2620| **2431**|
> |Obesity| 2.256| 0.063| **0.058**|
> |Power| 1.958| 1.571| **1.528**|
> |Protein| 1.724| 1.699| **1.692**|
> |STAR| 136.49| 130.85| **130.29**|
> |Superconductor| 1.929| 0.110| **0.087**|
> |Synthetic| 6.147| **5.767**| 5.771|
> |Wave| 19320| 4312| **4264**|
> |Wine| 0.382| 0.324| **0.320**|
> |Yacht| 3.485| 0.304| **0.300**|
> |*KNN-FI W-L*| 20-2| -| 2-20|
> |*IBUG W-L*| 21-1| 20-2| -|
>
> ### MACE/Sharpness
> |Dataset|KNN|KNN-FI|IBUG|
> | --- | --- | --- | --- |
> |Ames| **0.030**/34350| 0.090/**22243**| 0.049/24818|
> |Bike| **0.018**/42.894| 0.086/**1.198**| 0.069/1.924|
> |California| **0.016**/0.603| 0.089/**0.402**| 0.037/0.459|
> |Communities| 0.040/0.144| 0.051/0.142| **0.027**/**0.129**|
> |Concrete| **0.031**/9.093| 0.061/3.617| 0.081/**3.062**|
> |Energy| **0.050**/4.071| 0.189/0.382| 0.132/**0.254**|
> |Facebook| 0.115/**15.854**| 0.116/18.123| **0.080**/17.056|
> |Kin8nm| **0.056**/0.154| 0.104/**0.098**| 0.087/0.116|
> |Life| **0.032**/1.796| 0.073/**1.528**| 0.063/1.596|
> |MEPS| 0.065/**11.477**| 0.068/15.933| **0.064**/15.907|
> |MSD| 0.028/10.151| 0.083/9.940| **0.009**/**7.898**|
> |Naval| 0.126/0.002| 0.058/0.001| **0.043**/**0.000**|
> |News| 0.196/5180| 0.206/5167| **0.124**/**3431**|
> |Obesity| **0.022**/4.962| 0.115/0.179| 0.109/**0.111**|
> |Power| **0.015**/3.815| 0.041/3.429| 0.025/**3.148**|
> |Protein| **0.030**/**3.339**| 0.042/3.839| 0.038/3.751|
> |STAR| 0.027/247.67| 0.025/244.83| **0.021**/**235.55**|
> |Superconductor| **0.049**/5.286| 0.140/**0.129**| 0.079/0.250|
> |Synthetic| **0.010**/11.066| 0.013/10.497| 0.010/**10.301**|
> |Wave| **0.011**/42887| 0.191/6463| 0.206/**6372**|
> |Wine| **0.030**/0.702| 0.097/0.649| 0.093/**0.608**|
> |Yacht| 0.127/7.332| 0.123/**0.642**| **0.107**/0.795|
> |*KNN-FI W-L*| 3-19/19-3| -/-| 2-20/7-15|
> |*IBUG W-L*| 8-14/19-3| 20-2/15-7| -/-|

---

### Official Review · Reviewer_gTo6 · 2022-07-06

**Rating:** 6
**Confidence:** 4
**Soundness:** 3 good
**Presentation:** 3 good
**Contribution:** 3 good

**Summary:**

This paper proposes a new method to estimate data uncertainty in GBDT models. The proposed approach uses k nearest training elements to produce probabilistic predictions. Nearest elements are determined using the constructed ensemble: top instances are chosen based on the number of times they are in the same leaf with the test example. The method can be applied to any GBDT model after it is trained.

**Questions:**

1. Regarding the baselines, note that in CatBoost, there is a loss function called RMSEWithUncertainty, which, similarly to NGBoost, predicts mean and variance. I think that adding a comparison with this implementation is important. In this case, by comparing CatBoost+RMSEWithUncertainty with CatBoost+IBUG, one can see the effect of the proposed method without possible effects caused by differences between GBDT implementations.

2. Some training details are not clear to me. Namely, how exactly were the parameters tuned? Is it true that in Table 1 for NLL the parameters are tuned via NLL, for CRPS via CRPS, etc.? This seems to agree with Tables 5-6 in the supplementary, where we see different parameters for different measures. If this is true, does IBUG for RMSE coincide with the standard LightGBM algorithm?

3. There are some important questions that are not addressed in the main text, but the corresponding experiments are in the supplementary materials. For instance:
    - To speed up the inference, it is proposed to sample trees uniformly at random. However, for GBDT models, it is known that trees at the beginning of the ensemble and at the end are very different. Namely, the first trees give the most significant contribution, while the tree structures at the end are closer to random. Thus, a particular sampling strategy is important. This problem seems to be addressed in Appendix C.3 (but I am not sure what “in ascending order” means here). I expect a more detailed discussion about this in the main text.
    - Another important question is whether the proposed approach of computing the similarity is better than a naive KNN applied to the original features? This question is not discussed in the main text but seems to be addressed in Appendix B.3. I expect to see more details on this in the main text. Also, it is not discussed whether feature normalization is performed before applying KNN.

Minor:
- In line 605 of the supplementary materials, it is written that one-hot encoding is used for categorical variables. This can be suboptimal for some GBDT libraries, e.g., CatBoost has its own way of dealing with categorical variables.

The paper is, in general, well written, and I noticed only minor typos in the text:
- Line 87: “mean and variance is” (should be “are”), the same for lines 141 and 189
- Caption of Figure 2: “means” -> “mean”
- Line 252: “Table” -> “Figure"


**Limitations:**

Yes.

**Strengths And Weaknesses:**

Strengths:
- The method can work with any GBDT model and can be applied to any trained model at inference time.
- The reported results show that the proposed approach outperforms existing methods (NGBoost and PGBM) in terms of both RMSE and probabilistic evaluation measures.

Weaknesses
- The inference time can increase significantly.
- The paper does not compare with the existing probabilistic prediction for GBDT implemented in CatBoost (see details below in Questions).

---

> ### Author Response · Authors · 2022-08-02
> **Response to Reviewer gTo6 (Part 3)**
>
> **Q: What is the effect of different tree-sampling strategies?**
>
> **A:** We agree the sampling strategy can make a significant difference on the efficiency and efficacy of the probabilisitic predictions in IBUG. Sec. C.3 shows results for three different sampling strategies: *random*, *first-to-last*, and *last-to-first* (we will replace *ascending* with *first-to-last* and *descending* with *last-to-first*).
>
> We observe that sampling trees *last-to-first* generally requires sampling all trees in order to achieve the lowest NLL on the test set. In contrast, when sampling *first-to-last*, our results provide some evidence to the reviewer's comment that initial trees provide the most significant contributions; this is especially evident for the Naval, Protein, and Wine datasets in which sampling less than 5% of the trees results in the same or better NLL than when sampling all trees. However, on the Bike and Obesity datasets, *random* sampling achieves the lowest NLL with the smallest number of trees sampled (note the sharp "elbow" for these datasets in Figure 7a) out of all of the sampling strategies; thus, sampling a mixture of trees earlier and later in training is sometimes most beneficial. Thank you for your thoughtful comments, we will add this discussion to the main text and reference Figure 7 and Sec. C.3.
>
> **Q: Comparison to KNN applied to original features?**
>
> **A:** Yes, Sec. B.3 provides a comparison between IBUG and KNN applied to the original features. Overall, KNN was not competitive with IBUG in both point and probabilistic performance (Tables 7, 8, and 9). Due to space constraints, we omitted this comparison in the main text.
>
> Unfortunately, the results shown in Sec. B.3 does not use feature normalization before applying KNN. However, we have run additional experiments that *do* apply standard scaling before using KNN; the results are shown in the response to reviewer *VfPp*. Overall, we observe the same trends as in Sec B.3. Thank you for your suggestion, we will make sure to add a discussion of these results in the main text.
>
> **Typos.**
>
> **A:** Thank you for spotting these typos.

---

> > ### Comment · Reviewer_gTo6 · 2022-08-09
> > **Thanks for the reply**
> >
> > Thank you for the clarifications and additional experimental results. I increased my score.

---

> ### Author Response · Authors · 2022-08-02
> **Response to Reviewer gTo6 (Part 2)**
>
> ### MACE/Sharpness
> |Dataset|NGBoost|PGBM|CBU|IBUG-CB|CBU+IBUG-CB|
> | --- | --- | --- | --- | --- | --- |
> |Ames| 0.076/78290| **0.058**/**17692**| 0.088/20385| 0.090/20709| 0.077/19136|
> |Bike| 0.070/110.832| 0.098/1.884| **0.037**/2.159| 0.108/**1.224**| 0.056/1.581|
> |California| **0.012**/0.481| 0.053/**0.344**| 0.020/0.357| 0.095/0.371| 0.043/0.354|
> |Communities| **0.031**/0.125| 0.080/**0.123**| 0.044/0.125| 0.041/0.134| 0.049/0.128|
> |Concrete| **0.047**/6.308| 0.056/3.046| 0.101/3.288| 0.102/**2.494**| 0.061/2.765|
> |Energy| 0.158/1.152| 0.109/0.292| 0.057/0.365| 0.077/**0.270**| **0.054**/0.301|
> |Facebook| 0.095/9.186| 0.195/**3.830**| 0.100/10.515| **0.063**/18.165| 0.096/13.672|
> |Kin8nm| **0.021**/0.177| 0.049/0.162| 0.024/0.096| 0.135/**0.070**| 0.053/0.080|
> |Life| **0.041**/3.210| 0.064/**1.044**| 0.080/1.259| 0.142/1.276| 0.063/1.182|
> |MEPS| **0.031**/**8.693**| 0.073/10.107| 0.097/11.891| 0.086/17.381| 0.092/14.305|
> |MSD| **0.007**/7.743| 0.037/**7.435**| 0.011/8.097| 0.039/9.086| 0.030/8.491|
> |Naval| **0.033**/0.006| 0.237/0.001| 0.050/0.001| 0.061/**0.000**| 0.080/0.001|
> |News| 0.093/**2444**| 0.088/3284| **0.078**/3395| 0.207/4556| 0.085/3577|
> |Obesity| 0.063/1e20| 0.175/0.462| **0.021**/0.139| 0.102/**0.088**| 0.036/0.107|
> |Power| **0.017**/3.816| 0.033/2.643| 0.020/**2.367**| 0.031/3.320| 0.021/2.761|
> |Protein| 0.029/5.118| 0.088/**3.147**| 0.036/3.148| **0.015**/3.980| 0.044/3.500|
> |STAR| 0.029/247.19| 0.040/248.32| 0.024/**233.28**| 0.026/242.84| **0.023**/237.63|
> |Superconductor| 0.089/8.070| 0.082/**0.183**| 0.026/0.316| 0.166/0.242| **0.025**/0.253|
> |Synthetic| 0.010/14.39| 0.019/10.74| 0.013/**10.28**| **0.009**/10.41| 0.010/10.34|
> |Wave| 0.129/1e6| 0.019/6442| **0.008**/**4149**| 0.072/6581| 0.055/5096|
> |Wine| **0.017**/0.685| 0.083/**0.544**| 0.026/0.579| 0.090/0.643| 0.061/0.602|
> |Yacht| 0.118/4.085| 0.126/0.493| **0.083**/0.450| 0.116/0.479| 0.084/**0.439**|
> |*IBUG-CB W-L*| 7-15/17-5| 10-12/9-13| 5-17/7-15| -/-| 6-16/7-15|
> |*CBU+IBUG-CB W-L*| 8-14/17-5| 17-5/9-13| 9-13/11-11| 16-6/15-7| -/-|
>
> **Q: How are the hyperparameters tuned?**
>
> **A:** You are correct that for NLL in Table 1, the hyperparameters are optimized for NLL (shown in Table 5); for CRPS in Table 1, the selected hyperparameters are shown in Table 6. IBUG is a post-hoc approach applied to a given GBRT point predictor; thus, there are two sets of hyperparameters to tune, those for the point predictor (e.g., LightGBM, XGBoost, etc.), and those for IBUG ($k$, $\rho$, and $\gamma$/$\delta$). The hyperparameters for the point predictor are tuned based on a point-prediction metric such as RMSE, then the IBUG hyperparameters are tuned based on a probabilistic performance metric such as NLL or CRPS. Therefore, the LightGBM base model has the same selected hyperparameters in Tables 5 and 6 since the base model is tuned using RMSE, but IBUG has different selected hyperparameters since it is tuned using NLL and CRPS, respectively.

---

> ### Author Response · Authors · 2022-08-02
> **Response to Reviewer gTo6 (Part 1)**
>
> We thank the reviewer for their comments and suggestions, here we address major concerns.
>
> **Q: Comparison to CatBoost with loss function "RMSEWithUncertainty"?**
>
> **A:** Per the reviewer's suggestion, we compare IBUG-CB (IBUG-CatBoost) to CBU (CatBoost with loss function "RMSEWithUncertainty". The results are shown below (we also include NGBoost and PGBM for additional context); we observe IBUG and CBU achieve similar point and probabilisitic performance, and generally outperform both NGBoost and PGBM.
>
> These results use variance calibration (Sec. 3.2) which we note has a significant beneficial impact on CBU, especially for the MACE (mean absolute calibration error) results where the median MACE score improved by 3.2x when using variance calibration vs. without.
>
> We also note CBU can only model univariate gaussians or other distributions with location and scale, but IBUG can flexibly model any parametric distribution *and* nonparametric density estimators. To demonstrate this flexibility, we compute the NLL on the test sets of the MEPS and Wine datasets from Sec. 5.4 using CBU and IBUG-CB. On the MEPS dataset, CBU and IBUG-CB achieve an NLL of 3.781 +/- 0.040 and **-6.898 +/- 0.117**, respectively; on the Wine dataset, CBU and IBUG-CB achieve an NLL of 1.003 +/- 0.008 and **0.812 +/- 0.016**, respectively. In both cases, CBU modeled the output as a normal distribution while IBUG modeled the output as a Weibull distribution and using KDE (kernel density estimation) for the MEPS and Wine datasets, respectively.
>
> Since CBU and IBUG have similar performance, we experiment combining CBU and IBUG into an ensemble approach, in which their predictions (after variance calibration) are averaged. In the results below, we observe this hybrid approach performs surprisingly well, typically outperforming all other approaches, especially in terms of CRPS. Thus, when approaching a new dataset, using a combination of CBU and IBUG (potentially with different base models) may achieve the best performance. We thank the reviewer for their suggestion and we will add these results to the paper.
>
> ### RMSE
> |Dataset|NGBoost|PGBM|CBU|IBUG-CB|CBU+IBUG-CB|
> | --- | --- | --- | --- | --- | --- |
> |Ames| 71167| 22762| 21315| 21430| **21078**|
> |Bike| 47.165| 3.741| 3.043| 3.087| **2.949**|
> |California| 0.529| 0.440| 0.444| **0.424**| 0.427|
> |Communities| 0.139| 0.134| 0.131| 0.130| **0.130**|
> |Concrete| 6.121| 3.725| 3.511| 3.423| **3.420**|
> |Energy| 1.212| 0.329| 0.391| **0.279**| 0.320|
> |Facebook| 31.12| 20.32| 20.49| 20.38| **20.30**|
> |Kin8nm| 0.177| 0.108| 0.104| **0.086**| 0.091|
> |Life| 2.767| 1.703| 1.668| 1.655| **1.629**|
> |MEPS| 25.37| 23.87| 23.81| 23.81| **23.75**|
> |MSD| 9.290| 8.806| 8.742| 8.749| **8.725**|
> |Naval| 0.006| 0.000| 0.001| **0.000**| 0.000|
> |News| 11050| 11047| 10996| 11001| **10994**|
> |Obesity| 0.570| 0.181| 0.178| 0.167| **0.164**|
> |Power| 3.923| 2.979| 2.901| 2.904| **2.880**|
> |Protein| 4.963| **3.480**| 3.536| 3.532| 3.510|
> |STAR| 232.3| 230.6| 228.3| 228.0| **227.8**|
> |Superconductor| 7.344| **0.412**| 0.493| 0.444| 0.442|
> |Synthetic| 10.25| 10.21| 10.19| 10.19| **10.18**|
> |Wave| 1000905| 7901| 4623| 4635| **3835**|
> |Wine| 0.700| 0.607| 0.629| **0.602**| 0.603|
> |Yacht| 3.417| 0.695| 0.537| **0.465**| 0.471|
> |*IBUG-CB W-L*| 22-0| 19-3| 15-7| -| 6-16|
> |*CBU+IBUG-CB W-L*| 22-0| 20-2| 22-0| 16-6| -|
>
> ### CRPS
> |Dataset|NGBoost|PGBM|CBU|IBUG-CB|CBU+IBUG-CB|
> | --- | --- | --- | --- | --- | --- |
> |Ames| 37637| 10641| 10559| 9953| **9742**|
> |Bike| 12.481| 1.155| 0.833| 0.972| **0.776**|
> |California| 0.256| 0.219| 0.214| 0.210| **0.204**|
> |Communities| 0.066| 0.068| 0.065| 0.063| **0.063**|
> |Concrete| 3.266| 1.895| 1.744| 1.736| **1.659**|
> |Energy| 0.528| 0.163| 0.198| **0.142**| 0.158|
> |Facebook| 4.132| 3.626| 3.301| 3.192| **3.109**|
> |Kin8nm| 0.095| 0.071| 0.058| 0.052| **0.051**|
> |Life| 1.408| 0.819| 0.789| 0.801| **0.741**|
> |MEPS| **5.550**| 6.243| 6.140| 6.158| 6.042|
> |MSD| 4.523| 4.575| 4.360| 4.408| **4.345**|
> |Naval| 0.003| 0.000| 0.000| 0.000| **0.000**|
> |News| **2149**| 2328| 2293| 2510| 2324|
> |Obesity| 3e17| 0.113| 0.064| 0.058| **0.054**|
> |Power| 2.112| 1.534| 1.480| 1.535| **1.470**|
> |Protein| 2.672| 1.853| 1.798| 1.795| **1.753**|
> |STAR| 131.50| 131.00| 129.56| 129.47| **129.19**|
> |Superconductor| 2.429| **0.119**| 0.154| 0.151| 0.128|
> |Synthetic| 5.838| 5.774| 5.761| 5.762| **5.759**|
> |Wave| 570788| 3911| 2201| 2490| **1896**|
> |Wine| 0.385| 0.324| 0.338| 0.324| **0.323**|
> |Yacht| 0.967| 0.240| 0.229| 0.191| **0.187**|
> |*IBUG-CB W-L*| 20-2| 18-4| 14-8| -| 1-21|
> |*CBU+IBUG-CB W-L*| 20-2| 21-1| 21-1| 21-1| -|
>
> (results continued in part 2...)

---

### Official Review · Reviewer_E8u6 · 2022-07-12

**Rating:** 5
**Confidence:** 4
**Soundness:** 3 good
**Presentation:** 2 fair
**Contribution:** 3 good

**Summary:**

This paper addresses an interesting problem: uncertainty estimation for GBRT. The authors propose a $k$-nearest neighbors approach based on an affinity between the testing sample and training samples. To save computational time, sampling from trees is used. They prove by experiments that the proposed IBUG works better than NGBoost and PGBM, two recent gradient boosting algorithms for tree model uncertainty.

**Questions:**

1. What's the point prediction of IBUG? Is it the original GBRT prediction, or the $\mu_y$ obtained from the $k$-nearest neighbors set? If it is $\mu_y$, then is the prediction as good as the original GBRT prediction?

2. Authors did not compare the proposed method to any other methods beyond NGBoost and PGBM. Although there are very few works on uncertainty estimation for GBRT, one can come up with some naive ideas. For example, how about fixing the point predictions of GBRT as the mean of Gaussian distributions, and using a deep neural network to predict the variances of the Gaussian with maximum likelihood? What's the comparison of this naive solution to yours? I'm not familiar with PGBM, however, comparing only to PGBM (and NGBoost) is somewhat problematic.

3. In probabilistic forecasting, calibration error (with sharpness) is really a popular metric. Can you consider using it in experiments? ECE and sharpness can provide more comprehensive evaluations.

4. In calibrating variance, tuning $\gamma$ and $\delta$ on validation data is not a good solution. Can they be learned with a gradient descent?

5. The model performance versus varying $k$ is worthy to be studied. What is the suggested $k$ for the neighbors' set? Can it be fixed for all datasets? It seems $k$ varies greatly across datasets, as shown in the Appendix. This makes me confused, and there may be no guidelines on choosing $k$.

6. The minimum sample size in the leaf nodes is also a hyper-parameter in your model. What are its effects on the affinity calculation and the uncertainty estimation?

7. The writing can be improved, especially the mathematical expressions.
In Equation (1), the sum starts from $t=1$, not $i=1$.
Equation (3) and Equation (4) need to be modified.

**Limitations:**

No potential negative societal impact.

**Strengths And Weaknesses:**

Strengths:
The idea has some similarities to existing works, such as distance-based conformal prediction. The way to calculate the distance (or the affinity) is new. So this paper shows a new idea, and the authors show the usefulness of the model. Comparisons to PGBM verify the benefits of such a GBRT uncertainty model.

Weaknesses:
I have some concerns about the technical issues. Please see the list in the question section.

---

> ### Author Response · Authors · 2022-08-02
> **Response to Reviewer E8u6**
>
> Thank you for your comments and suggestions, here we address major concerns.
>
> **Q: What's the point prediction of IBUG? Is it the original GBRT prediction, or is it obtained from the nearest-neighbors set? If it is, then is the prediction as good as the original GBRT prediction?**
>
> **A:** Yes, you are correct that the point prediction of IBUG is the original GBRT prediction (Sec. 3.1). In preliminary experiments, we tested using the *k*-nearest neighbors to model the conditional mean, but we found using the original GBRT prediction achieves better point predictions and subsequently better probabilistic predictions.
>
> **Q: No comparisons beyond NGBoost and PGBM? Can you use a neural network to estimate the variance?**
>
> **A:** NGBoost and PGBM are recent methods that achieve state-of-the-art results on tabular probabilistic regression problems; our approach is conceptually simple, easy to implement, and is generally competitive with these approaches. Using a neural network to estimate the variance adds a great deal of complexity and may require careful tuning of the architecture and hyperparameters, and may not work well for problems with limited data.
>
> Additional KNN and random baseline results are in Sec. B.3 of the Appendix. However, we also add comparisons to *three* more methods based on other reviewers' feedback. Specifically, we compare against: (1) a KNN method that uses feature importance from the GBRT to reduce the dimensionality and avoid the curse of dimensionality (see response to *VfPp*); (2) BART, a popular but expensive sampling based approach (see response to *TLnP*); and (3) CBU, CatBoost model using the loss function "RMSEWithUncertainty" (see response to *gTo6*).
>
> **Q: In probabilistic forecasting, calibration error (with sharpness) is really a popular metric. Can you consider using it in experiments? ECE and sharpness can provide more comprehensive evaluations.**
>
> **A:** Thank you for the suggestion, we have added a combination of MACE (mean absolute calibration error, equivalent to ECE)/sharpness to our probabilistic performance evaluation. We show results for these metrics in our responses to the other reviewers, and provide new analyses and insights into existing state-of-the-art methods.
>
> **Tuning hyperparameters $\gamma$, $\delta$, and $k$**.
>
> **A:** Using a validation set is a valid and very common method for tuning hyperparameters. Yes, the optimal value of $k$ does depend on the dataset, so we follow the standard KNN practice of choosing it based on validation data. Variance calibration (tuning $\gamma$ and $\delta$) is done for all methods as a final step, after all other hyperparameters are tuned. This calibration substantially improves several baseline methods as well as IBUG.
>
> **Q: The minimum sample size in the leaf nodes is also a hyper-parameter in your model. What are its effects on the affinity calculation and the uncertainty estimation?**
>
> **A:** Minimum sample size in leaf nodes is a hyper-parameter of the GBDT learning algorithms. We tune this hyper-parameter to minimize RMSE, and then apply IBUG, which works with the tree ensemble as-is. In our experiments, we considered the hyper-parameter values [1, 20] (Sec. B.2); "1" was selected by 12 datasets and "20" by 8 datasets. In terms of efficiency, the affinity computation is faster for leaf nodes with a small number of examples, and is only affected by very large leaves (we observe LightGBM tends to produce large leaves in Sec. B.5). In terms of probabilistic performance, we observe no correlation between minimum leaf size and relative IBUG performance; for datasets where the minimum leaf size is "1", IBUG has better mean NLL (CRPS) scores on 9/12 (11/12) and 8/12 (10/12) datasets when compared against NGBoost and PGBM, respectively; when the minimum leaf size is "20", IBUG achieves better mean NLL (CRPS) on 10/10 (9/10) and 8/10 (7/10) (Table 1).
>
> **Mathematical expressions.**
>
> **A:** Thank you for spotting the typo in Equation (1). As for Equations (3) and (4), we will revise those sections for better clarity.

---

### Author Response · Authors · 2022-08-02
**General Response**

We thank all the reviewers for their valuable comments and suggestions. We are encouraged reviewers find this an interesting and important problem (*E8u6*, *VfPp*) in which our approach is clear, simple, and sound (*gTo6*, *VfPp*, *TLnP*), yet flexible and effective (*gTo6*, *TLnP*) with an extensive empirical evaluation (*VfPp*) and a high degree of engineering (*TLnP*).

During this short response period, we have focused first on the additional experiments requested by the reviewers, including additional baselines and metrics. Thank you for the suggestions -- we believe these additional comparisons will make the paper stronger. We will integrate them into a revised version of the paper ASAP. Additional baselines include:
1. *KNN-FI*, a KNN method that uses feature importance from the GBRT to reduce the dimensionality and avoid the curse of dimensionality (see response to reviewer *VfPp*).
2. *BART*, a popular but expensive sampling-based approach (see response to reviewer *TLnP*).
2. *CBU*, a CatBoost model using the loss function “RMSEWithUncertainty” (see response to reviewer *gTo6*).

Additional metrics include MACE (mean absolute calibration error, equivalent to ECE) and sharpness (lower is better for both). Sharpness quantifies the average of the standard deviations and thus does not depend on the actual ground-truth label; therefore, MACE and sharpness are shown together, with better methods having both low calibration error and low sharpness scores. For all metrics, scores are averaged over the 20 random folds for each dataset. Additional reviewer-specific concerns are shown in response to each reviewer.

---

### Meta-Review · Area_Chair_LDZD · 2022-08-28

**Recommendation:** Accept
**Confidence:** Less certain

**Metareview:**

This paper presents a method  for extending any GBRT point predictor to produce probabilistic predictions such that the aleatoric uncertainty can be quantified. It computes a nonparametric distribution around a prediction using the k NNs where the distance is measured by a kernel that is similar to the random forest kernel. The paper is well written and easy to read. All of reviewers agree that it is a simple practical method that is well engineered. But all the techniques used in this system are existing ones, so that its technical novelty is limited. During the discussion period, I had more than a few communications with reviewers. One one hand, there were some concerns on the limited novelty, which I also agree with. In fact, this concern became more notable in the discussion. On the other hand, a strength is in its simplicity, practicability, and its excellence in engineering and design, how to critically evaluate alternative approaches, and how to design experiments that evaluate those approaches”. A few things that I would like the authors to consider in their future submissions include: (1) the method is applied to quantify only aleatoric uncertainty, which should be clearly mentioned in an earlier place in the paper, since these days we observe a few interesting methods for quantifying the predictive uncertainty (that is both aleatoric and episdemic uncertainty); (2) a kernel which is similar to the random forest kernel is used as a distance metric. Unlike RF, GBRT construct trees with small depth, so that it is expected that many instances fall in the same leaf. The behavior might be different from the case of RF. Despite a concern on the limited novelty, most of reviewers feel that this work can be accepted, so I recommend it for acceptance.


**Award:**

No

---

### Decision · Program_Chairs · 2022-09-14

Accept